# Changes in Diarrhea Score, Nutrient Digestibility, Zinc Utilization, Intestinal Immune Profiles, and Fecal Microbiome in Weaned Piglets by Different Forms of Zinc

**DOI:** 10.3390/ani11051356

**Published:** 2021-05-11

**Authors:** Han-Jin Oh, Yei-Ju Park, Jae Hyoung Cho, Min-Ho Song, Bon-Hee Gu, Won Yun, Ji-Hwan Lee, Ji-Seon An, Yong-Ju Kim, Jun-Soeng Lee, Sheena Kim, Hyeri Kim, Eun Sol Kim, Byoung-Kon Lee, Byeong-Woo Kim, Hyeun Bum Kim, Jin-Ho Cho, Myung-Hoo Kim

**Affiliations:** 1Department of Animal Sciences, Chungbuk National University, Cheongju 286-44, Korea; dhgkswls17@naver.com (H.-J.O.); adidats@naver.com (W.Y.); junenet123@naver.com (J.-H.L.); ajs6@daum.net (J.-S.A.); xormakzm@naver.com (Y.-J.K.); abyz314@naver.com (J.-S.L.); bklee0418@cherrybro.com (B.-K.L.); 2Department of Animal Sciences, Pusan National University, Miryang 50463, Korea; zx7777777@pusan.ac.kr (Y.-J.P.); kimbw@pusan.ac.kr (B.-W.K.); 3Department of Animal Resource, and Science, Dankook University, Cheonan 311-16, Korea; jhcho5216@gmail.com (J.H.C.); sheen915@gmail.com (S.K.); hyely26@gmail.com (H.K.); essol0430@gmail.com (E.S.K.); 4Division of Animal and Dairy Science, Chungnam National University, Daejeon 341-34, Korea; mhsong@cnu.ac.kr; 5Life and Industry Convergence Research Institute, Pusan National University, Mirayng 50463, Korea; g.bonhee@gmail.com

**Keywords:** ZnO alternative, piglets, digestibility, immune, microbiome

## Abstract

**Simple Summary:**

Piglets, especially at the weaning stage, are highly susceptible to various diseases due to an incomplete immune system development and stress responses. Post-weaning diarrhea has a significant impact on piglet growth rate and mortality, resulting in economic losses to the swine industry. Zinc oxide (ZnO) is widely used as a weaning diet supplement in the swine industry to prevent diarrheal diseases and promote immune system development. Despite the recently demonstrated beneficial effects of ZnO, many efforts have been made to reduce its excessive use in piglets owing to environmental pollution and toxic effects on tissues; thus, the need for an effective alternative ZnO form, which promotes zinc utilization, has been gaining attention. However, we do not completely understand the mode of action of ZnO alternatives or the amount required to exert positive effects on weaned piglets. Therefore, we conducted this study to evaluate the effects of different forms of ZnO alternatives (ZnO chelate with glycine (chelate-ZnO) and nanoparticle-sized ZnO (nano-ZnO)) on diarrhea score, nutrient digestibility, zinc utilization, intestinal immune profiles, and fecal microflora on piglets, together with a comparison of the standard ZnO treatment. We found that 200 ppm Nano-ZnO had similar positive effects on weaned piglets compared with 2500 ppm ZnO and therefore is a promising alternative to ZnO.

**Abstract:**

Twenty weaned piglets with initial body weight of 6.83 ± 0.33 kg (21 day of age, LYD) were randomly assigned to four treatments for a two-week feeding trial to determine the effects of different dietary zinc on nutrient digestibility, intestinal health, and microbiome of weaned piglets. The dietary treatments included a negative control (CON), standard ZnO (ZnO, 2500 ppm), zinc chelate with glycine (Chelate-ZnO, 200 ppm), and nanoparticle-sized ZnO (Nano-ZnO, 200 ppm). At 0 to 1 week, the diarrhea score was decreased in the CON group compared with the ZnO, Chelate-ZnO, and Nano-ZnO group. In overall period, the ZnO and Nano-ZnO groups exhibited improved diarrhea scores compared to the CON group. The apparent total tract digestibility of dry matter and gross energy was the lowest in the CON group after one week. Compared to the ZnO group, the chelate-ZnO group exhibited higher proportion of T-bet+ and FoxP3+ T cells and the nano-ZnO group had higher numbers of RORgt+ and GATA3+ T cells in the mesenteric lymph nodes. ZnO group increased IL-6 and IL-8 levels in the colon tissues and these positive effects were observed in both chelate ZnO and nano-ZnO groups with lower level. The 16S rRNA gene analysis showed that the relative abundance of *Prevotella* was higher in the ZnO-treated groups than in the CON group and that of *Succinivibrio* was the highest in the nano-ZnO group. The relative abundance of *Lactobacillus* increased in the ZnO group. In conclusion, low nano-ZnO levels have similar effects on nutrient digestibility, fecal microflora, and intestinal immune profiles in weaning pigs; thus, nano-ZnO could be used as a ZnO alternative for promoting ZnO utilization and intestinal immunity.

## 1. Introduction

Post-weaning diarrhea (PWD) leads to a decrease in the growth rate of pigs and represents a major cause of mortality in weaned piglets, thus resulting in significant economic losses to the swine industry [1]. In the post-weaning period, the piglets are subjected to many stress factors, such as dietary changes, histological changes in the small intestine, removal from the sow, mixing of pigs from different pens, or adaptation to a new environment [2]. These factors negatively affect the intestinal environment of weaned pigs [1,2,3]. PWD is commonly caused by the proliferation of enterotoxigenic *Escherichia coli* (ETEC). Zinc oxide (ZnO) is often supplemented to weaning diets to prevent PWD. Feeding pharmacological levels of zinc (2000–4000 ppm as ZnO) to weaned pigs has been shown to immediately prevent recurring diarrhea and improve growth rate following weaning [4]. Zinc is a well-known immunomodulatory nutrient and has diverse functional roles in promoting immune system development and regulating immune responses. For example, zinc treatment promotes immune system development, including that of B cells, monocytes, and granulocytes [5]. Zinc also regulates the production of immunomodulatory molecules from immune cells. For instance, zinc promotes secretion of chemo-attractants and expression of TNFα, IL-6, and IL-1β in immune cells [6,7,8]. ZnO administration regulates both innate and adaptive immune responses in piglets. ZnO treatment upregulates defensin 3 expression in the jejunum of piglets following *E. coli* exposure [9]. Zinc treatment also activates adaptive immunity in weaned piglets; ZnO supplementation promotes CD3+T cell proliferation in the blood of piglets [10]. In weaned piglets, zinc treatment had differential impact on the mRNA expression of transcription factors of T helper 1 cell (T-bet) and regulatory T cell (FoxP3) in the intestine in a time dependent manner. Short term zinc treatment (one week) increased the T-bet expression, however long term feeding (four weeks) decreased the expression of T-bet and increased FoxP3 expression [11]. While zinc supplementation decreases Th17 cells in the spleen of mice [12], zinc treatment during the weaning period increases anti-inflammatory T cells called regulatory T cells (Treg) in whole blood [10]. Pei et al. (2018) [13] and Kolubert et al. (2018) [10] reported that exposure to high doses of zinc induces anti-inflammatory cytokines and disrupts natural killer cell (NK cell) activation. Collectively, ZnO is a powerful immunomodulatory nutrient for piglets, but its effectiveness seems to depend on the levels of ZnO. Zinc administration is known to induce changes in the gut microbiome; ZnO supplementation reduces the abundance of opportunistic pathogens including Campylobacterales, Enterobacteriaceae, and Escherichia, resulting in improved gut integrity [14,15,16]. Despite the recently demonstrated beneficial effects of ZnO usage, many efforts have been made to reduce its excessive use in piglets as high levels of ZnO in the diets of weaned pigs are mostly excreted, resulting in markedly high concentration of zinc in the soil [17]. The European Union has decided to phase out the use of ZnO in swine feed by 2022. In the meantime, the formulation of a low-zinc piglet diet has been proposed to decrease the amount of zinc excreted. Such a formulation would include a source of zinc with high bioavailability. In previous studies, organic zinc—especially zinc combined with an amino acid—has shown a higher bioavailability than the commonly used inorganic zinc [4,17,18,19,20]. Chelating zinc prevents its precipitation within the small intestine and improves absorption via the peptide or amino acid transport system [21,22]. Schlegel et al. (2010) [23] reported that zinc chelate bound to glycine is more stable than ZnSO_4_ and is advantageous for zinc absorption because of the presence of phytase in the gut. Nanoparticles are characterized by a large surface area, nanoscale size, and high catalytic effect and surface activity of the particles [24]. Nanoparticle-sized ZnO (Nano-ZnO) has shown effective antimicrobial activity against *E. coli* and has an effect similar to that of high-dose ZnO on intestinal morphology, suggesting that Nano-ZnO has the potential to replace ZnO in swine diets [25,26]. The nutrient requirement of swine (NRC; 2012) [27] reported that the dietary requirements of zinc are only 80–100 mg/kg in weaned piglets. Therefore, we conducted an in vivo study to evaluate the effects of alternative forms of ZnO with lower level, which may serve as alternatives to ZnO, on the diarrhea score, nutrient digestibility, zinc utilization, intestinal immune profiles, and fecal microbiome of weaned piglets. This study has significant implications to expand our understanding that how various biological factors, which regulated by zinc are connected to each other to prevent diarrheal disease in weaned piglets.

## 2. Materials and Methods

### 2.1. Animals, Facilities, and Dietary Treatments

The experimental protocol for this study was reviewed and approved by the Institutional Animal Care and Use Committee of Chungbuk National University, Cheongju, Korea (approval no. CBNUA-1421-20-02). A total of 20 weaned piglets (Duroc × Landrace × Yorkshire; 21 day of old) were allotted to a completely randomized block design. The pigs (average initial body weight of 6.43 ± 0.33 kg) were individually placed in 45 × 55 × 45 cm stainless steel metabolism cages in an environmentally controlled room (30 ± 1 °C). There were one pig treatment in a cage and five replicate cage per treatments. The dietary treatments consisted of CON (negative control; no additional added zinc oxide in diet), ZnO (positive control; CON + 2500 ppm/kg zinc oxide), chelate-ZnO (CON + 200 ppm/kg of zinc chelate with glycine), and nano-ZnO (CON + 200 ppm/kg nanoparticle-sized zinc oxide; Table 1). All diets were formulated to meet or exceed the NRC [27]. The daily feed allowance was adjusted to 2.7 times the maintenance requirement for digestible energy (DE; 2.7 × 110 kcal of DE/kg BW^0.75^). This allowance was divided into two equal parts, and the piglets were fed at 08:30 and 17:30 each day. The diets were mixed with water in a 1:1 ratio (*w/w*) before feeding. Water was provided ad libitum through a drinking nipple. We individually weighed the pigs at the beginning of each period and recorded the amount of feed supplied and any residual feed quantity for each period. 

### 2.2. Growth Performance and Diarrhea Scores

The individual BWs and feed intake were documented at the end of 7 and 14 days, the average daily gain (ADG), average daily feed intake (ADFI), and feed efficiency (G:F) was determined. The subjective diarrhea scores were individually recorded at 09:00 and 18:00 by the same person on days 0 to 14 post weaning. The diarrhea score was assigned as follows: 0, diarrhea; 1, sloppy feces; 2 normal feces; and 3, well-formed feces. Scores were calculated as the average diarrhea score for each period (0 to 7 days; 7 to 14 days; overall period, 0 to 14 days) per group by summing the average daily diarrhea scores of each pigs.

### 2.3. Sample Collections

The first experimental period consisted of a four-day adaptation period, followed by a three-day collection period to collect feces. The feed was the same during the second experimental period as that in the first experimental period. We set a four-day feces collection period and alternated the feeding time between the day of slaughter and the previous two days so that pigs could be slaughtered within the designated time (Figure 1). Fecal samples were collected directly from the rectum of the 12 piglets before (week 0) and after dietary supplementation (week 2) (randomly selected three piglets per group: CON, ZnO, chelate-ZnO, and nano-ZnO). 16S rRNA gene library preparation was performed as previously described [28], and MiSeq sequencing of 16SrRNA gene amplicons was conducted using the Illumina MiSeq platform at Macrogen Inc. (Seoul, Republic of Korea). The entire spleen was weighed. One-third of the mesenteric lymph nodes (MLNs) were removed and stored in a phosphate-buffer solution (PBS). The intestinal tract was incised 20 cm from the end of the ileum along the abdominal gland to remove it. The jejunum and proximal colon were stored in 10% neutral buffered formalin (NBF; Sigma-Aldrich, St. Louis, MO, USA) and TRIzol (Ambion, Austin, TX, USA). The intestinal tissues in 10% NBF were stored at RT, and the tissues were transported onto dry ice and stored at −80 °C in TRIzol before post-processing. The fecal samples were frozen in a plastic bag. The ileal digesta was then freeze-dried. The samples were finely crushed and stored at −20 °C to measure zinc content. Feces were immediately collected as they appeared in the metabolism cages. They were stored in a freezer at −20 °C until analysis. Fecal samples were dried at 70 °C for 72 h in a forced-air oven and ground through a 1-mm screen. The samples were thoroughly mixed, and a subsample was collected for chemical analysis. Before euthanasia, blood samples from the piglets were collected into 10 mL K_2_EDTA tubes (BD Bioscience, NJ, USA) for PBMC isolation, 0.5 mL K_3_EDTA tubes (Greiner Bio One, Kremsmunster, Austria) for CBC analysis, and non-heparinized tubes for serum and vacuum tubes containing K_3_EDTA (Becton, Dickinson and Co., Franklin Lakes, NJ, USA) for whole blood collection. The blood sample tubes were stored on ice until analysis. After collection, serum samples were centrifuged (3000× *g*) at 4 °C for 20 min. 

### 2.4. Chemical Analysis for Diet and Feces

Diets and feces were analyzed for dry matter (DM), crude protein (CP), and gross energy (GE) using AOAC methods [29]. We analyzed the GE of the diets and feces using an adiabatic oxygen bomb calorimeter (Parr Instruments, Moline, IL, USA). Diets, feces, and ileal digesta samples were wet digested using nitric-perchloric acid and then diluted with deionized distilled water for mineral analysis. The concentration of zinc was analyzed using UV absorption spectrophotometry (UV-1201; Shimadzu, Tokyo, Japan). We calculated the apparent total tract digestibility (ATTD) of DM, CP, GE, and Zn, as well as the average daily mineral intake, using the following equations: (ATTD lrb%) = ([DI × NID − OF × NIF]/[DI × NID]) × 100; Average daily mineral intake = ADFI × MD; DI is the DM intake (g), NID is the nutrient content (DM, CP, GE, and Zn) of diet on a DM basis; OF is the output of feces (g); and NIF is the nutrient content of the feces on a DM basis. MD is the Zn content in the diet.

### 2.5. Fecal Microbiome Analysis

Mothur software was used to remove low-quality sequences from the 16S rRNA gene sequences [30]. Briefly, sequences that did not match the PCR primers were discarded from the demultiplexed sequence reads. To minimize the effects of random sequencing errors, sequences containing ambiguous base calls and sequences with a length of less than 100 bp were trimmed. Chimeric sequences were eliminated using the UCHIME algorithm implemented in Mothur. With an OTU definition at an identity cutoff of 97%, de novo operational taxonomic unit (OTU) clustering was conducted using the Quantitative Insights into Microbial Ecology (QIIME) software package (version 1.9.1) [31]. The naïve Bayesian RDP classifier and the Greengenes reference database were used for the taxonomic assignment of the sequences. Beta-diversity was measured using unweighted UniFrac distance metrics using QIIME, which considers community membership (presence or absence of OTUs) [32]. Principal coordinate analysis (PCoA) plots were generated based on unweighted UniFrac distance metrics.

### 2.6. Complete Blood Count Analysis

The red blood cell (RBC), white blood cell (WBC), and lymphocyte levels in the whole blood were determined using an automatic blood analyzer (ADVIA 120, Bayer, Tarrytown, NY, USA). Immunoglobulin G (IgG) and immunoglobulin M (IgM) levels were determined using an automatic biochemistry blood analyzer (Hitachi 747; Hitachi, Tokyo, Japan). The zinc concentration in the blood was determined according to the method described by Hill et al. [33]. Blood samples in the 0.5 mL K_3_EDTA tubes were analyzed using Vetscan HM5 (ABAXIS, Union City, CA, USA). Monocytes and neutrophils were counted in the whole blood sample cells. The percentages of lymphocytes, monocytes, and neutrophils in the WBCs were analyzed. 

### 2.7. Flow Cytometry Analysis

MLNs were ground using a 40-µm strainer and resuspended in PBS and ACK buffer. MLN cells were stained with a live/dead fixable aqua dead cell stain kit (Invitrogen, Carlsbad, CA, USA) and anti-pig CD4a (BD Bioscience). Then, the cells were fixed with an eBioscience^TM^ FoxP3/Transcription factor staining buffer set (Invitrogen), and the transcription factors were stained with anti-mouse T-bet (Invitrogen), anti-mouse RORgt (Invitrogen), anti-mouse FoxP3 (Invitrogen), and anti-mouse GATA3 (BioLegend, San Diego, CA, USA). The stained cells were evaluated using a CANTO II flow cytometer (BD Biosciences), and the data were analyzed using FlowJo software v10.7.1 (Tree Star Inc., Ashland, OR, USA).

### 2.8. Histological Analysis

Intestinal tissues were stored in 10% NBF for two weeks at room temperature. The samples were cut to a width of 2 cm. After dehydration and dealcoholization, the intestine samples were embedded in paraffin and sectioned. The slides were deparaffinized, rehydrated, and stained with H&E. The samples were examined using an Olympus IX51 inverted phase-contrast microscope.

### 2.9. Quantitative RT-PCR

The MLNs were ground on a 40-µm strainer, and the RBCs were removed by ACK buffer treatment. The MLN cells were seeded at a density of 1 × 10^6^ cells/well in a six-well plate with LPS (5 µg/mL; Sigma) and incubated at 37 °C and 5% CO_2_ for 4 h. After incubation, the MLNs were harvested and stored in TRIzol at −80 °C. Before RNA isolation, the intestinal tissues were homogenized using Bioprep-6 (Bioand, Gyeonggi-do, Korea) with 3-mm metal beads, the samples were centrifuged (10,000× *g*) at 4 °C for 5 min, and the supernatant was transferred to fresh tubes. Samples in TRIzol were incubated for 5 min at room temperature, and 200 µL of chloroform was added per mL of TRIzol to the initial tubes; the tubes were then vortexed for 10 s, left to sit for 2 min at room temperature, and then centrifuged (10,000× *g*) at 4 °C for 20 min. The upper aqueous phase was transferred into a fresh tube; 0.5–1 mL isopropyl alcohol was added, and the samples were gently mixed by shaking. After incubation for 10 min at room temperature and centrifugation (10,000× *g*) at 4 °C for 10 min, the supernatant was removed; the resulting RNA pellet was washed with 75% ethanol before being stored in DEPC water (Invitrogen). cDNA was synthesized using Accupower RT PreMix (Bioneer, Daejeon, Korea). Real-time qRT-PCR was performed using QuantStudio1 (Applied Biosystems, Foster City, CA, USA) and Solog^TM^ h-Taq DNA Polymerase (SolGent, Daejeon, Korea). The reaction conditions were as follows: 50 °C for 10 min, 95 °C for 5 min, 95 °C for 15 s, and 60 °C for 30 s (40 cycles), followed by melting curve analysis. Primers are described at Table 2. Expression levels were calculated using the ΔΔC_t_ method after correcting for differences in PCR efficiencies and expressed relative to the CON levels.

### 2.10. Statistical Analysis

Orthogonal contrasts (control vs. other treatments; ZnO vs. chelate-ZnO; ZnO vs. nano-ZnO; chelate-ZnO vs. nano-ZnO) were used to compare the possible relationships between the treatments using the PROC procedure general linear model (GLM) of SAS (Statistical Analysis System 9.4, SAS Institute, Cary, NC, USA). For the ADG, ADFI, G:F, diarrhea score, nutrient digestibility, zinc excretion, ATTD of zinc, AID of zinc and blood profiles were calculated within trial, phase, and treatment combination. These averages were considered as outcome variables and, within each phase, performed with analysis of variance (ANOVA) using PROC GLM with SAS. The immunological parameters were calculated using GraphPad Prism version 7.00 for Windows (GraphPad Software, San Diego, CA, USA, www.graphpad.com, accessed on 3 February 2021). Data from the CBC analysis, spleen weight measurement, and qRT-PCR were assessed using a one-way ANOVA with GLM procedures. A two-way ANOVA was used to confirm the taxonomic analysis data. Differences among treatment means were determined using Tukey’s test. Flow cytometry data were analyzed using an unpaired *t*-test. For microbiome analysis, a two-sided Welch’s *t*-test in the Statistical Analysis of Metagenomic Profiles (STAMP) software v2.1.3 [34] was used to identify significant differences in the relative abundance of microbial taxa among the four groups. Analysis of similarities (ANOSIM) was used to determine whether the microbial compositions between the four groups were significantly different using QIIME and were based on the unweighted UniFrac distance metrics. Statistical significance was set at *p* < 0.05.

## 3. Results

### 3.1. Effects of Different Forms of Zinc-Containing Diets on Growth Performance and Diarrhea Incidence in Piglets

At 0 to 7 days, pigs fed the control diet (CON) had lower (*p* < 0.01; contrast *p* < 0.05) diarrhea score than did the pigs fed the ZnO, chelate-ZnO, and nano-ZnO diets (Table 3). There was no difference in diarrhea scores among ZnO, chelate-ZnO, and nano-ZnO diets. At 7 to 14 days, pigs fed the ZnO diet exhibited higher (*p* < 0.01; contrast *p* = 0.001, 0.032, 0.001) diarrhea scores than did the pigs fed the nano-ZnO diet. However, pigs fed the control diet had lower (*p* < 0.05; contrast *p* < 0.05) diarrhea scores than did the pigs that were fed the other treatments. During the overall experimental period (0 to 14 days), pigs fed the CON diet had lower (*p* < 0.01; contrast *p* < 0.05) diarrhea score than did the pigs fed the ZnO and nano-ZnO diets. Moreover, pigs fed the chelate-ZnO diet had lower (*p* < 0.01; contrast *p* < 0.05) diarrhea score than did the pigs fed the ZnO diet. 

### 3.2. Nutrient Digestibility and Zinc Utilization in Piglets Fed Different Forms of Zinc

The ATTD of DM and GE was significantly (*p* < 0.05; contrast *p* < 0.05) decreased in pigs of the CON treatment group compared with that in those of the other treatment groups at one week (Table 4). The energy excretion in the feces of the ZnO treatment group was significantly decreased compared with that of the chelate-ZnO treatment group at two weeks. In the same period, the ATTD of GE was significantly decreased (*p* < 0.05; contrast *p* < 0.05) in the CON and chelate-ZnO treatment groups compared with that in the ZnO treatment group. Pigs fed the ZnO diet displayed significantly higher (*p* < 0.05) zinc intake than those fed the chelate-ZnO and nano-ZnO diets when CON treatment was excluded; however, there were no significant differences (*p* > 0.05) between the zinc uptake of pigs in the chelate-ZnO and nano-ZnO treatment groups at 1 and 2 weeks (Table 5). Compared with those fed the ZnO diet, pigs fed the chelate-ZnO and nano-ZnO diets had lower (*p* < 0.05; contrast *p* < 0.05) zinc excretion in feces and higher ATTD of zinc at 1 and 2 weeks. Compared with those fed the chelate-ZnO diet, pigs fed with nano-ZnO exhibited significantly lower (*p* < 0.05) zinc AID and zinc ATTD. The blood concentration of zinc was significantly increased (*p* < 0.05; contrast, *p* < 0.05) in the ZnO treatment group compared with that in the other treatment groups (Table 6).

### 3.3. Blood Immune Profiles and Zinc Levels in Piglets Fed Different Forms of Zinc 

To determine the effect of different forms of dietary zinc on systemic immunity, we examined the blood immune profiles of piglets via CBC analysis. CBC analysis indicated that the number of WBCs in whole blood and the percentage of lymphocytes, monocytes, and neutrophils did not change significantly between groups fed different forms of zinc-containing diet (Table 6). In addition, the concentrations of blood immunoglobulin (Ig), IgG and IgM, were not affected by the different dietary treatments. To classify immune organ development in the weaned piglets, spleen weight was measured after two weeks of supplementation with ZnO. Spleen weight was not significantly different among all treatments (Figure 2). Collectively, ZnO and its alternatives used in this study had no significant effect on blood immune profiles. 

### 3.4. Changes in Intestinal Morphology in Piglets Fed Different Forms of Zinc 

To determine whether different forms of dietary zinc affected intestinal structural development in weaned piglets, we performed histological analysis of the small intestine (jejunum) and the large intestine (proximal colon) in piglets treated with different forms of dietary zinc. ZnO and its alternatives had no significant effect on intestinal tissue morphology. Villus length and crypt depth did not significantly differ in piglets treated with different forms of zinc, and no pathological damage to villi was found in any of the groups (Figure 3).

### 3.5. T Cell Subset and Cytokine Expression in Gut Lymph Node of Piglets Fed Different Forms of Zinc

To determine the effects of different forms of zinc on the intestinal immunity of the piglets, we further examined CD4+ T cell populations in MLNs by flow cytometry. The proportion of total CD4+ T cell population did not differ between the ZnO treatment group or any type of ZnO alternative groups (Figure 4A,B). At 2 weeks of feeding, the 2500 ppm ZnO diet induced a small decrease in CD4+ T cell subsets (T-bet+, RORgt+, GATA3+, FoxP3+ T cells) in the MLN, but the difference was not significant (Figure 4C,D). Notably, ZnO alternatives did influence subsets of CD4+ T cell populations in MLNs. The percentages of T-bet+CD4+ T cells (Th1) and FoxP3+CD4+ T cells (Treg) were higher in the chelate-ZnO group than in the ZnO group (Figure 4C,D). The percentages of GATA3+CD4+ T cells (Th2) and RORγt+CD4+ (Th17) T cells were significantly higher in the nano-ZnO group than in the ZnO group. The nano-ZnO group had a numerically higher proportion of Th1 and Treg cells than did the ZnO group (Figure 4C,D). To examine the functional response of immune cells in the intestine, we further examined the mRNA expression of cytokines in immune cells from MLNs. The suspended MLN cells were stimulated with LPS for 4 h, and cytokine expression was measured using qRT-PCR. We examined the major cytokines (IL-8, IL-6, IFNγ, IL-1β, IL-10, IL-17A, and IL-22); however, we did not observe significant changes in cytokine expression following LPS stimulation in this study (Figure 5). These findings indicate that ZnO and its alternatives have minor effects on immunity in the gut lymph node. 

### 3.6. mRNA Expression of T Cell Transcription Factors and Cytokine Expression in the Colon Tissues of Piglets Fed Different Forms of Zinc

We examined the expression of T cell transcription factors, including TBX21, GATA3, RORC, and FoxP3, in colon tissue using qRT-PCR. However, we did not observe any significant differences among the dietary treatment groups (Figure 6). Additionally, expression of cytokines—including IL-8, IL-2, IFNγ, IL-6, IL-1β, IL-10, IL-17A, IL-22, IL-12, and IL-4—was examined in the colon tissues. We observed significant changes in the expression of some cytokines in the colon tissues; the mRNA expression levels of IL-6 and IL-8 were significantly increased in the piglets of the ZnO treatment group compared to the levels of those in the CON group (Figure 7). Piglets in both the chelate-ZnO and nano-ZnO diets also showed similar effects on IL-6 and IL-8 expression levels in the colon tissues. Furthermore, the piglets in the Chelate-ZnO group exhibited higher expression levels of IL-10 and IFNγ in the colon tissues than did those in the CON or ZnO group (Figure 7).

### 3.7. mRNA Expression of Tight Junction Proteins and Antimicrobial Peptides in the Colon Tissue of Piglets Fed Different Forms of Zinc

We examined the gut barrier and antimicrobial functions in the colon tissue of the piglets. First, the expression of tight junction proteins (ZO-1, OLCN1, and CLDN4) was examined by qRT-PCR. We found that the expression of ZO-1, OLCN1, and CLDN4 was not influenced by any zinc treatment (Figure 8A). We also examined the expression of porcine beta defensin 1 (pBD1) and regenerating islet-derived protein 3 gamma (REGⅢγ), an antimicrobial peptide. Dietary zinc treatment did not affect pBD1 expression, but alternative forms of ZnO treatments tended to increase the expression level of REGIIIγ in colon tissue (Figure 8B). Collectively, the expression of barrier and antimicrobial function-related genes did not differ among the dietary treatment groups.

### 3.8. Changes in the Fecal Microbiomes of Piglets by the Supplementation with Different Forms of Zinc

The *Lactobacillus* concentration in the feces samples was significantly decreased (*p* < 0.05; contrast *p* < 0.05) in the CON treatment group compared with that in the other treatment groups (Table 6). Sequencing of the 16S rRNA genes in the fecal samples produced a total of 2,781,610 reads after quality filtering, with a mean sequence number of 231,800 ± 20,953 reads per sample. ANOSIM of the unweighted UniFrac distances indicated that each group did not cluster significantly (*p* > 0.05, data not shown). The analysis of the two groups before and after dietary treatment (week 0, week 2) revealed significant differences, suggesting that the microbiota of pigs in the week 0 and week 2 groups were significantly different (*p* < 0.05). The unweighted UniFrac PCoA plot visually confirmed the distinct separation of microbial communities between the groups (Figure 9). Comparisons of the relative abundances of the gut microbiota compositions between the four treatment groups at the phylum and genus levels are shown in Figure 10. At the phylum level, the bacterial sequences from the CON samples were composed predominantly of the phyla Firmicutes (51.49%), Bacteroidetes (24.16%), Proteobacteria (16.19%), and four other phyla that collectively comprised 8.16% of the total sequences analyzed (Figure 9). The microbiota of the pigs in the ZnO group consisted largely of phyla Firmicutes (49.44%), Bacteroidetes (29.52%), Proteobacteria (14.43%), and four other phyla, which collectively comprised 6.61% of the total sequences analyzed (Figure 10A). In the samples of the Chelate-ZnO group, Firmicutes (43.09%), Bacteroidetes (34.40%), and Proteobacteria (14.51%) were predominant, and the remaining four phyla comprised 8.00% of the total sequences (Figure 10A). In the samples of the Nano-ZnO group, Firmicutes (36.61%), Bacteroidetes (35.46%), and Proteobacteria (22.07%) were predominant, whereas the other four phyla comprised 5.86% of the total sequences analyzed (Figure 10A).

At the genus level, *Prevotella* was the most enriched genus in all fecal samples (Figure 10B). Its relative abundance increased significantly in the ZnO, chelate-ZnO, and nano-ZnO (38.04%, 49.07%, and 41.32%) groups compared to that in the CON group (29.08%; Figure 11A). The relative abundance of *Succinivibrio* significantly increased from an average of 24.78% in the nano-ZnO group, whereas its abundance in the ZnO and chelate-ZnO groups decreased (7.15% and 13.28%, respectively) compared to that in the CON group (14.49%; Figure 11B). Even though *Lactobacillus* represented the most abundant genus (10.75%) in the ZnO group, its abundance in the chelate-ZnO and nano-ZnO groups decreased significantly (1.05% and 2.25%, respectively), even more than that in the CON group (3.82%; Figure 11C). 

## 4. Discussion

Zinc is an essential mineral with various enzymatic roles. It improves immunity and the composition of the body structure. It also helps in the development of the gastrointestinal tract in pigs, preventing diarrhea, and thus affecting growth [17]. The natural zinc content of feedstuff is insufficient for pigs, and zinc is usually added in an inorganic form, such as ZnO or ZnSO_4_ [35]. The ZnO form has low reactivity and bioavailability, and the ZnSO_4_ form is hygroscopic and reacts with rapid ions to form free radicals to accelerate the breakdown of fatty acids, vitamins, and other nutrients in the feed [36]. In addition, to prevent diarrhea during the weaning phase, zinc that cannot be absorbed is discharged in the feces, which presents a major environmental problem [17]. For weaning piglets, supplementation of high dose zinc usually shows beneficial effects at 14 days post weaning treatment [4,10]. Thus, we treated piglets with different types of zinc for two weeks post weaning in this study. 2500 ppm ZnO treatment decreased diarrhea incidence in weaned piglets, as reported by several other studies. More importantly, dietary supplementation with a low dose (200 ppm) of nano-ZnO presented a similar diarrhea score to that of a high dose (2500 ppm) of ZnO during the experimental period. Similarly, 300 ppm/kg of nano-ZnO in the weaning diet was found to significantly decrease the diarrhea incidence in weaned piglets, and the effect was similar to that of 3000 ppm/kg of zinc oxide in diets during 14-day experiments [37]. Sun et al. (2019) [38] reported that 400–600 ppm/kg of nano-ZnO in weaning diets significantly decreases the incidence of diarrhea, and the supplementation of 600 ppm/kg of nano-ZnO is comparable to that of the addition of 2000 ppm/kg of ZnO in the weaning diet. The antidiarrheal mechanism of high zinc supplementation is not well understood [10]. Additionally, in vitro study, zinc supplementation at high dosages leads to negative effects, and alterations are recognized that are similar to those of zinc deficiency [8]. Moerover, in weaned piglets, high dosage of zinc administration decreased the number of lymphocytes in ileal lymph node [39]. Furthermore, supplementing weaned piglets with nano-ZnO did not have any effect on the growth performance of the weaned piglets. Even at high levels of nano-ZnO, the residual doses did not increase [13]. Excessive feed intake is one of the main causes of diarrhea in piglets, as it contributes to intestinal inflammation and adversely affects villi height and crypt depth [40]. In addition, the proliferation of bacteria, such as *E. coli*, is promoted by undigested proteins, leading to damage to the epithelium by toxins and bacteria [41]. However, in the present study, the incidence of diarrhea in piglets was relatively low because we had limited the supply of diets owing to metabolic experiments; thus, we believe that further experimental studies should be conducted. When investigating nutrient digestibility, we found that pigs fed with 2500 ppm/kg of ZnO or 200 ppm/kg of nano-ZnO and chelate-ZnO had a higher ATTD of DM and GE than pigs fed with control diet at one week. There was no significant difference between the ZnO and nano-ZnO treatments at two weeks. Hu et al. [37] reported that supplementation of diets with ZnO can improve the activation of digestive enzymes in the small intestine and pancreatic tissue, thereby improving the digestibility of nutrients. Other studies have reported that small intestine morphology improved due to the pharmacological supplementation with ZnO, resulting in improved nutrient digestibility and diarrhea score [42,43]. Our results indicated that supplementation with nano-ZnO could work similarly to that of ZnO pharmacological supplementation. Bękowski et al. [44] reported a linear correlation between nutrient and energy utilization with increasing amounts of dietary supplemented nano-ZnO. The pigs in the chelate-ZnO group had a higher digestibility than the that of nano-ZnO and ZnO treatments. Star et al. [45] reported that the bioavailability of organic zinc was higher than in inorganic zinc sulfate. When comparing the relative biological value of organic zinc, considering zinc sulfate to be 1, the ratio was found to be 1.64. Li et al. [43] reported that the bioavailability of zinc was affected by the form of zinc, and in contrast to inorganic zinc, organic zinc is absorbed through the amino acid system or the peptide transport system and shows a high digestibility. These findings are similar to those of the present study; in this study, pigs in the nano-ZnO treatment had a higher zinc digestibility than that in the ZnO treatment. This result indicates that the nano-ZnO had a larger surface area and particle surface activity, which is similar to the results of other studies [24,46,47]. However, in contrast to the results of other studies [13,46], our findings did not show a significant difference in zinc content in the blood across the different treatments, except for the ZnO treatment. In particular, despite the high ATTD and AID of zinc in the chelate-ZnO treatment, the zinc content in the blood did not differ. Many researchers have reported that dietary pharmacological supplementation of ZnO improves the microbial composition in the intestine, thereby reducing pathogenic microorganisms and increasing beneficial bacteria [13,14,16]. However, Lei & Kim [43] and Li et al. [48] reported that coated or pharmacological supplementation of ZnO had no effect on *Lactobacillus* or *E. coli* concentrations in the small intestine and cecum. In our study, there was no significant difference in the concentration of *E. coli* in the feces. The concentration of *Lactobacillus* was significantly higher in the pigs in the ZnO, chelate-ZnO, and nano-ZnO groups than in the CON group. Similarly, Cho et al. [49] reported that fecal *Lactobacillus* concentrations were higher at 14 and 42 days after finishing the weaning diets with 3000 ppm/kg of ZnO and 200–300 ppm/kg of modified ZnO added, and there was no significant difference in fecal *E. coli* concentrations. Therefore, regardless of the form, it is suggested that if zinc is supplied to the diet, it will have the same beneficial effects in the intestine. Through supplementation with zinc, the gut microbiome showed a significant increase in the genera *Prevotella*, *Succinivibrio*, and *Lactobacillus*. *Prevotella* is known to degrade polysaccharides in plant cell walls to short-chain fatty acids (SCFAs), which are used as an energy source by facilitating the degradation of carbohydrates and proteins and protecting the animal from gut inflammation [28,50,51]. *Succinivibrio* has been identified as a high-cellulose-degrading bacterium with the ability to digest complex carbohydrates, which may help piglets to digest feed better [52]. Generally, *Lactobacillus* is known to play a role as a carbohydrate-utilizing bacterium with genes associated with carbohydrate transport and utilization [53,54,55]. In the genus *Lactobacillus*, only the piglets on the ZnO group showed a significantly higher abundance compared to that of the other groups, while the piglets in the chelate-ZnO and nano-ZnO groups showed lower abundances than those in the CON group. There was a higher abundance of other carbohydrate-utilizing bacteria in these two groups, while the abundance of *Lactobacillus* was low. In these studies, the higher the abundance of *Prevotella*, *Succinivibrio*, and *Lactobacillus*, the better the adaptation to the feeds in piglets. The intestinal tissue is a highly complex organ that includes the intestinal epithelium layer, lamina propria, mucus, microorganisms, and the local immune system. Piglet intestines undergo structural development, which confers a fortified gut barrier function. Zinc administration influences the development of the intestinal tissue structure and immune system. Many studies have reported that the administration of zinc oxide for over three weeks changes intestinal morphology [13,56,57]. Some studies also reported that after a two-week experiment, the morphology of the jejunum differs between piglets treated with a high dose of ZnO (2000–2250 ppm) compared to those in the negative control or treated with a low dose of different types of zinc oxide [58,59]. However, in this study, the morphologies of the jejunum and colon were not altered by different forms of zinc supplementation. The gut epithelial barrier is important for preventing unnecessary inflammation caused by dietary antigens and pathogen infection in the intestine. This gut barrier function is mediated by major tight junction proteins such as ZO-1, OCLN, and Claudin 4. However, in this study, the mRNA expression of these tight junction proteins in the colon was not altered by different forms of zinc oxide. Similarly, Shen et al. [58] reported that ZO-1 expression did not change between high dose (2250 ppm) and low dose (250 ppm) administration of zinc in the jejunal mucosa. Additionally, Zhang et al. [36] reported that the expression levels of ZO-1, occludin, and Claudin 4 were not altered by the dose of ZnO in the ileal mucosa. Collectively, in this study, zinc oxide treatment of weaning piglets had no significant impact on intestinal structure and barrier function. Zinc is a functional nutrient for regulating the immune system, and zinc deficiency affects the development of the immune system [60]. Zinc deficiency is responsible for thymic atrophy; therefore, zinc is essential for immature T cell differentiation [8]. The immune systems of piglets develop from the fetal stage to the neonatal period. At approximately three weeks of pregnancy, the development of the fetal thymus and spleen is initiated. At birth, intestinal physiological reactions to commensal microflora occur, and this response is characterized by gut and systemic immune organ development [61]. During the weaning period (2–4 weeks of age), young animals have the maximum number of novel antigens throughout their life [62]. At this point, the intestinal immune system is shaped by commensals and harmless dietary antigens, which is a critical step in maintaining gut immune homeostasis. In this study, we examined the effect of zinc on systemic immune system development via secondary lymphoid tissue development and blood immune profiles. The spleen is a secondary lymphoid organ and contains a large number of immune cells. Moreover, the spleen is involved in lymphocyte production and migration [63]. Some pathogens and cellular debris are removed by macrophages in the spleen, and some adaptive immune responses are initiated by antigen presenting cells (APCs) that recognize and target antigens [64]. In the present study, we demonstrated that spleen weight was not affected by the type of zinc oxide supplemented in the diets of weaned piglets. Similarly, Van Heugten et al. [65] reported that spleen weights in weaned pigs that were administered zinc sources between 0 and 160 ppm of zinc sources did not change. However, in rats, low level of zinc intake increases spleen weights [66]. In addition, the concentrations of white blood cells and the population of neutrophils and monocytes were not affected by the different forms of zinc oxide. Similar to the results of our study, Kloubert et al. [10] found that the total WBC count and the ratio of neutrophils to monocytes were not influenced by the different dietary zinc levels. Sun et al. [38] reported that when 400–600 ppm of nano-ZnO is supplied, IgM and IgG levels increase, which indicates that the immune function to protect against pathogenic viruses and microorganisms improves. In our study, numerical values were higher in the nano-ZnO and ZnO treatments than in the other groups; however, there was no significant difference between the groups. Zinc is a functional nutrient for regulating the innate and adaptive immune systems, and zinc deficiency affects the function of the immune system [60]. The innate immune system is the first defense system in animals with microbial infections. Unlike the adaptive immune system, innate immune cells react to a broad range of pathogens, so they initiate inflammatory responses earlier than adaptive immune cells. Innate immunity protects animals at an early stage of pathogen infection and is important for the effective induction of adaptive immunity. Antimicrobial peptides are part of the innate immune system and protect against bacterial infection by directly killing microbial pathogens. The gut is constantly exposed to various microbes, and intestinal epithelial cells secrete various AMPs to protect them from microbial invasion. In innate immunity, zinc deficiency leads to impairment of the function of monocytes, macrophages, neutrophils, and NK cells [8]. Zinc deficiency impairs the phagocytosis of neutrophils and macrophages, NK cell activity, and neutrophil recruitment and chemotaxis [67,68]. Some antimicrobial peptides decrease when exposed to zinc deficiency [69]. Zinc deficiency also affects the adaptive immune system. For example, zinc deficiency affects adaptive immune cell survival, proliferation, and differentiation [6]. Furthermore, zinc is essential for immature T-cell differentiation [8]. Normally, zinc deficiency leads to a decrease in CD4+ T cells and Th1/Th2 imbalance by impairing Th1 polarization [8,70]. Therefore, zinc is an essential nutrient for the normal development and functioning of the immune system in animals. Zinc administration has a significant impact on both innate and adaptive immune functions. Zinc supplementation induces cytokine production, including IL-1, IL-6, and TNF-α, in mononuclear cells in vitro, and induces neutrophil chemotaxis [71]. In NK cells, zinc supplementation increases the differentiation of NK cells and their cytotoxicity [6]. Although we did not investigate innate immune cells and their functions in this study, we examined antimicrobial peptide expression, which is a component of innate immunity. Supplementation with chelate-ZnO and nano-ZnO numerically increased REGIIIg expression in the colon tissues of the experimental piglets. Antimicrobial peptides, such as human beta defensin 2 (HBD2), HBD3, REGⅢβ, and REGⅢγ, are primarily produced by intestinal epithelial cells in response to innate cytokines IL-18 and IL-22. Antimicrobial peptides control the levels of pathogens in the lumen of the intestine and prevent bacterial infiltration into the gut. Zinc treatment has also been shown to induce the antimicrobial peptide LL-37 in human intestinal epithelial cells (Caco-2 cells) [72].

The adaptive immune system is composed of specialized immune cells. Through pathogen memory, adaptive immune cells react faster from the second infection of a specific pathogen. These cells eliminate pathogens directly or indirectly. T and B cells are the major adaptive immune cells. CD4+ T cells, called helper T cells, play critical roles in controlling immune responses by producing various cytokines. CD4+ helper T cells consist of four major subsets, Th1, Th2, Th17, and Treg, classified by their transcription factor or cytokine expression. Each CD4+ T cell subset has a distinct function in the immune response. The intestinal immune system development in piglets is described as follows: at birth, the structures and cells involved in the mucosal immune response are initially absent in pigs. Immediately after birth, the intestinal epithelium and lamina propria have very few lymphocytes, and immunological structures are not clear. Within two weeks of birth, intestinal T lymphocytes are populated, but these cells do not express CD4 or CD8 surface markers. In the next two weeks of life, CD4+ T cells appear in the intestinal mucosa, primarily in the lamina propria. At five weeks of age, the intestinal epithelium became colonized by CD8+ T cells. The intestinal architecture is developed to be comparable to that of a mature animal intestine at seven weeks of age. The MLN is the largest lymph node in the body. The APCs present intestinal antigens to T lymphocytes in MLNs, where immune responses are initiated and shaped [73]. Naïve T cells develop from the thymus and migrate into the lymph nodes, such as the MLN, and then differentiate into T cell subsets by increasing the expression of the transcription factors T-bet, GATA3, RORgt, or FoxP3. Then, they migrate to small or large intestine and perform their immunological functions. Hence, in this study, we examined changes in the MLN CD4+ T cell subset, Th1, Th2, Th17, and Treg in weaned piglets fed different forms of zinc. Interestingly, we observed that the total number of CD4+ T cells and subsets of CD4+ T cell subsets in the MLN differed across the different forms of dietary zinc oxide. Although the difference was not significant, ZnO treatment numerically decreased the proportion of CD4+ T cell subsets. Some studies have reported that high doses of zinc oxide impair lymphocyte function. In humans, excessive zinc intake can impair lymphocyte proliferation and immunosuppression [74]. In several in vitro studies, high zinc concentration in media has been shown to inhibit T cell function and has the potential to suppress the allogenic immune response [75]. Another study reported that a high concentration of zinc led to the impairment of all T cell functions with cytotoxic effects [67]. As a high dose of ZnO is able to dampen T cell responses, an appropriate and tested concentration of ZnO needs to be applied to weaned piglets. Pigs fed alternative forms of ZnO had a higher proportion of T cell subsets than that of the ZnO group. FoxP3+ and T-bet+ T cells were significantly higher in chelate-ZnO diet-fed pigs than in ZnO diet-fed pigs. Treg cells (CD4+FoxP3+) limit the development and progression of the immune response and suppress the development of the inflammatory response. Th1 cells (CD4+T-bet+) mainly act against viral and bacterial pathogens. An increased level Treg cells alongside with increased Th1 cells was not an common reaction because they have opposite functions. However, in some studies, anti-inflammatory T cells and pro-inflammatory T cells were increased at the same time, this reaction is important to maintain immune-homeostasis [76,77]. When pro-inflammatory T cells were differentiated from naïve CD4+ T cells by immunological challenges such as antigen exposure, the population of Treg cells were also increased to maintain immune-homeostasis by suppressing inflammatory responses. Nano-ZnO diet-treated pigs had higher populations of RORγt+ and GATA3+ T cells compared to that of the ZnO diet-treated pigs. Th17 cells (CD4+RORγt+) react with specific fungi and extracellular bacteria and contribute to immunological homeostasis. Th2 cells (CD4+GATA3+) defend against helminth infection and contribute to allergic inflammation. When we considered that these pigs did not fully develop their intestinal immune systems that may have abundant naïve CD4+ T cells but not differentiated CD4+ T cell subsets [78], a low dose of chelate-ZnO and Nano-ZnO may help in the differentiation of CD4+ T cells, unlike the effects of high doses of ZnO in weaned pigs. However, we believe that the observed effects from alternative forms of zinc were very mild, as pigs fed Chelate-ZnO and Nano-ZnO had similar proportions of T cell subsets compared with those fed the CON diet. Also, Janczyk et al. (2018) reported that there were no significant differences in major cytokine expressions in LPS stimulated MLN from weaned piglets treated by different types of zinc oxide [39]. This study reported that high dosage of zinc (2500 ppm of zinc) have positive effect on anti-salmonella function first week after infection, but from 7–10 days after infection, positive effects were disappeared. To fully understand the effect of zinc on intestinal T cell development and response, an experimentally induced infection model and examination of antigen specific-immune responses needs to be applied to perform in vivo studies with conditions of T cell expansion.

Additionally, we observed differential cytokine expression in the colon between the different forms of zinc oxide used in this study. IL-6 and IL-8 expression levels were significantly increased in pigs in the ZnO treatment group. Chelate-ZnO- and nano-ZnO-treated pigs also had similar levels of these cytokines compared with the ZnO-treated pigs. IL-6 is released from macrophages, endothelial cells, and T cells, and contributes to host defense through hematopoiesis and immune reactions. IL-8 is a chemotactic factor of neutrophils and T cells and is involved in neutrophil activation. Several types of immune cells, such as monocytes and macrophages, release IL-8. Some studies have reported similar results: ZnO-treated piglets have higher IL-8 expression levels in the colon compared to that of non-treated groups [56], and IL-6 serum levels have been shown to be higher in pigs treated with 3000 ppm ZnO than in those in the non-treated group [13]. Both IL-6 and IL-8 are generally considered as a pro-inflammatory cytokine that may reduce growth performance of adult animals. Undoubtedly, unnecessary, or overreaction of inflammatory response leads to negative impact on animals [79,80]. However, these cytokines are also important for inducing immune responses to fight against foreign materials. For example, IL-6 contributes T cell differentiation and IL-8 involved in immune cell proliferation and tissue remodeling [81,82,83]. When we consider that intestinal immune system of piglets used in this study was not fully developed, increased expressions of IL-6 and IL-8 represents promoted gut immunity, which may contribute to development of intestinal immune system during early age of piglets. In summary, high-dose ZnO treatment led to increased IL-8 and IL-6 expression in the colon of weaning piglets, and low doses of chelate-ZnO and nano-ZnO also showed similar effects on the expression of these cytokines in the intestinal tissue. In chelate-ZnO-treated pigs, IL-10, and IFN-γ expression increased compared to that of CON and ZnO pigs in this study. IL-10 is an anti-inflammatory cytokine that inhibits pro-inflammatory responses. Macrophages, dendritic cells, and Treg cells are the major sources of IL-10. IFN-γ promotes macrophage activation, B cell isotype switching, and T cell differentiation, and is released by Th1 cells, NK cells, and group 1 ILCs. These findings are in line with increased levels of MLN Tregs and Th1 in chelate-ZnO-treated pigs observed in this study. Collectively, the expression level of some cytokines in the colon tissue increased after 2500 ppm ZnO administration, and similar effects were observed in weaned pigs fed Chelate-ZnO and Nano-ZnO diets at low levels (200 ppm). In this study, we examined changes in the intestinal immune system induced by different forms of dietary zinc oxide. Most studies on the immunological effects of zinc in pigs have analyzed the immune profiles of serum or blood after zinc treatment. As zinc is a micronutrient that is absorbed through the intestine and weaned piglets can easily contract diarrheal diseases, examination of intestinal immunity provides critical information for the understanding of zinc effects.

## 5. Conclusions

The supplementation of nanoparticle-sized ZnO did showed similar effects on the diarrhea score, intestinal immunity, nutrient digestibility, and fecal microflora as compared to the pharmacological supplementation of ZnO in weaned pigs. A lower dosage of nanoparticle-sized ZnO could potentially replace ZnO, which causes environmental problems at the pharmacological addition level. Nanoparticle-sized ZnO could enhance beneficial genera that contribute to the degradation of polysaccharides and complex carbohydrates of plant cell walls into SCFAs, which can be used as an energy source and prevent gut inflammation, resulting in improved growth performance.

## Figures and Tables

**Figure 1 animals-11-01356-f001:**
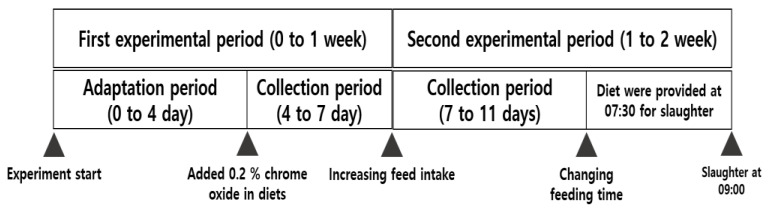
Schematic diagram of experimental design.

**Figure 2 animals-11-01356-f002:**
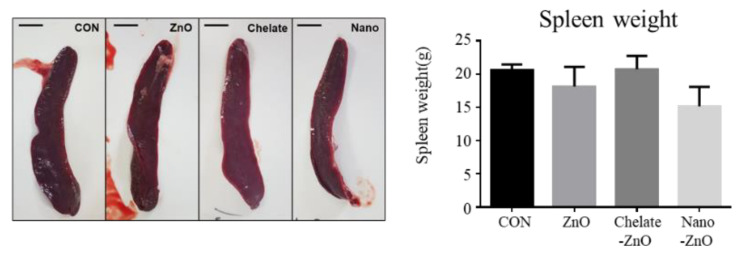
Spleen weights in weaned piglets fed alternative form of zinc oxide. Weaned piglets were administered different form of zinc oxide for two weeks. The control piglets were fed diets with no zinc oxide. The figure displays gross pictures and average weights of the spleen from four dietary pig groups (control, CON; ZnO, positive control; chelate-ZnO, zinc chelate with glycerin; and nano-ZnO, nanoparticle-sized zinc oxide). Data are represented as mean ± SD. Values were statistically analyzed by one-way ANOVA with Tukey’s multiple comparison test.

**Figure 3 animals-11-01356-f003:**
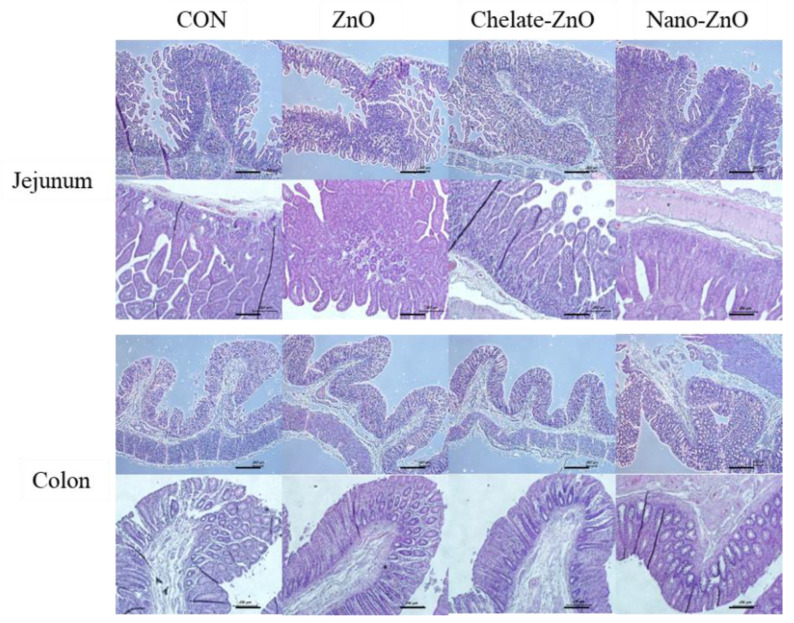
Histological analysis of the intestinal morphology of weaned piglets fed alternative form of zinc oxide. Weaned piglets were given different type of zinc oxide for two weeks. Control piglets were fed without zinc oxide. Figures displays morphology of jejunum and colon tissue from pigs in four dietary treatments. Scale bar = 100 μM.

**Figure 4 animals-11-01356-f004:**
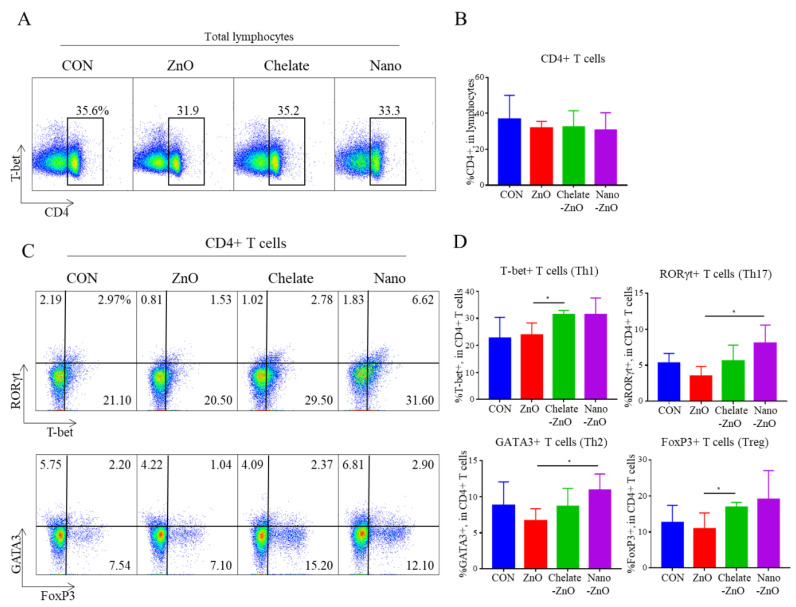
Population of CD4+ T cell subsets in mesenteric lymph node (MLN) of weaned piglets fed alternative form of zinc oxide. Forward scatter and side scatter were used to identify lymphocytes among live cells. Figures represent dot plots and graphs of CD4+ T cell (**A**,**B**) and its subsets (**C**,**D**) in piglets fed different forms of zinc oxide. CD4+ T cell subsets were identified by using the transcription factors T-bet, GATA3, RORγt, and FoxP3; T-bet, RORγt, and FoxP3 were gated simultaneously. Data are represented as mean ± SD. Values were statistically analyzed by unpaired *t*-test. * *p* < 0.05.

**Figure 5 animals-11-01356-f005:**
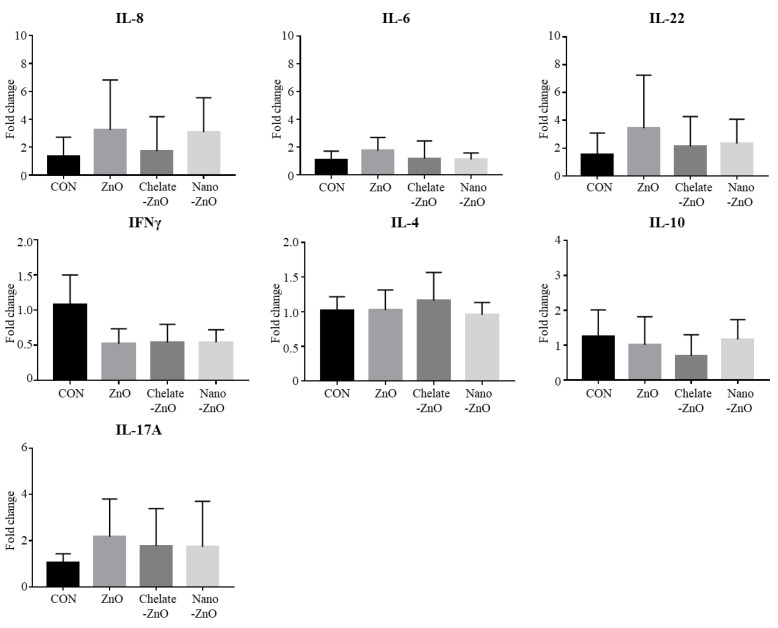
Major cytokine expression in the mesenteric lymph nodes (MLN) of weaned piglets fed different forms of zinc oxide. IL-8, IL-6, IL-22, IFNγ, IL-4, IL-10, and IL-17A gene expressions in the MLN were examined in piglets weaned on different forms of zinc oxide by using qRT-PCR. MLN cells were stimulated with 5 ug/mL of LPS for four hours on 37 °C and 5% CO_2_ condition. mRNA expression levels of the cytokines in each group were calculated based on that of the negative control group (CON). All fold change values were normalized by GAPDH. Data are represented as mean ± SD. Values were statistically analyzed by one-way ANOVA with Tukey’s multiple comparison test.

**Figure 6 animals-11-01356-f006:**
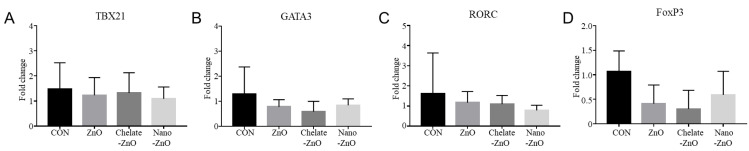
Gene expression of T cell subsets in the colons of weaned pigs fed alternative forms of zinc. T cell subset changes in the colons of piglets exposed to different forms of zinc oxide over two weeks were examined by determining T cell transcription factor expression using qRT-PCR. mRNA expression levels of TBX21 (**A**), GATA3 (**B**), RORC (**C**), and FoxP3 (**D**) in each experimental group were calculated based on that of the negative control group (CON). All fold change values were normalized by GAPDH. Data are represented as mean ± SD. Values were statistically analyzed by one-way ANOVA with Tukey’s multiple comparison test.

**Figure 7 animals-11-01356-f007:**
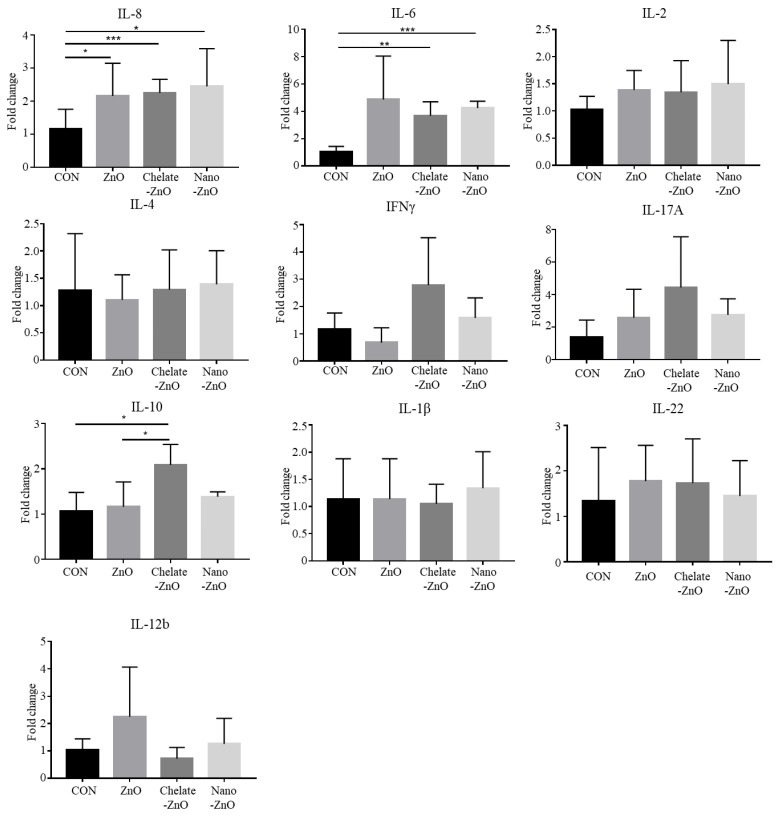
Cytokine expression in the colon tissues of weaned piglets fed different forms of zinc oxide. IL-8, IL-2, IL-4, IFNγ, IL-17A, IL-10, IL1â, IL-22, and IL-12b gene expression was examined using qRT-PCR in the colon tissues of weaning piglets that were administered different forms of dietary zinc oxide. mRNA expression levels of the cytokines in each group were calculated based on that of the negative control group (CON). All fold change values were normalized by GAPDH. Data are represented as mean ± SD. Values were statistically analyzed by one-way ANOVA with Tukey’s multiple comparison test. * *p* < 0.05; ** *p* < 0.005; *** *p* < 0.001.

**Figure 8 animals-11-01356-f008:**
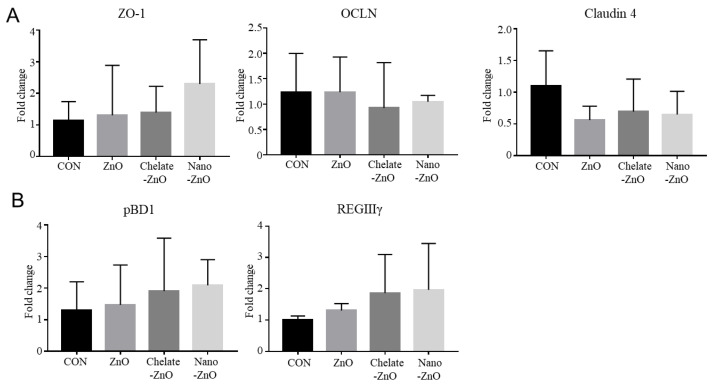
Barrier function in the colon tissues of weaned piglets fed different forms of zinc oxide. The gene expression of barrier functioning (**A**) and antimicrobial peptides (**B**) in colon tissues are shown. The mRNA expressions of the tight-junction related genes, ZO-1, OCLN, and CLDN4, were analyzed. The mRNA expressions of the antimicrobial peptide genes pBD1 and REGIIIγ were analyzed. Gene expression levels of each group were calculated based on that of the negative control group (CON). All fold change values were normalized by GAPDH. Data are represented as mean ± SD. Values were statistically analyzed by one-way ANOVA with Tukey’s multiple comparison test.

**Figure 9 animals-11-01356-f009:**
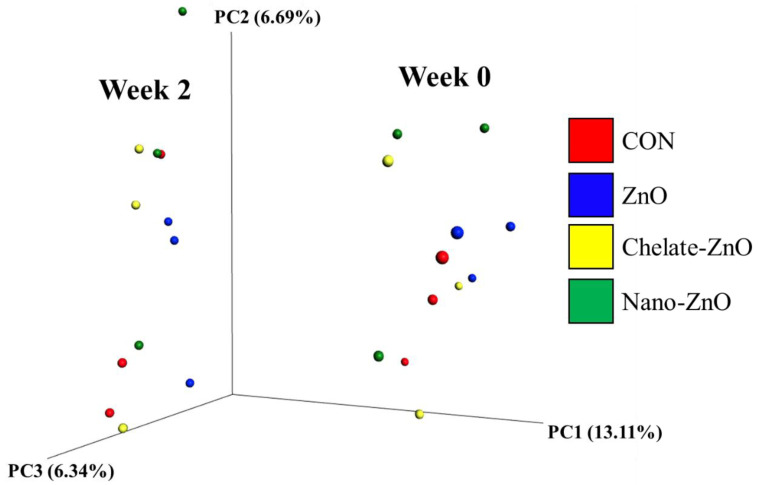
Principal coordinates analysis (PCoA) plots of ZnO treatment (CON; red. ZnO; blue, chelate-ZnO; yellow, nano-ZnO; green) in weeks. Week 0 (**right**) and week 2 (**left**) group clustered significantly based on unweighted UniFrac distance metrics (*p* < 0.05, R: 0.498346).

**Figure 10 animals-11-01356-f010:**
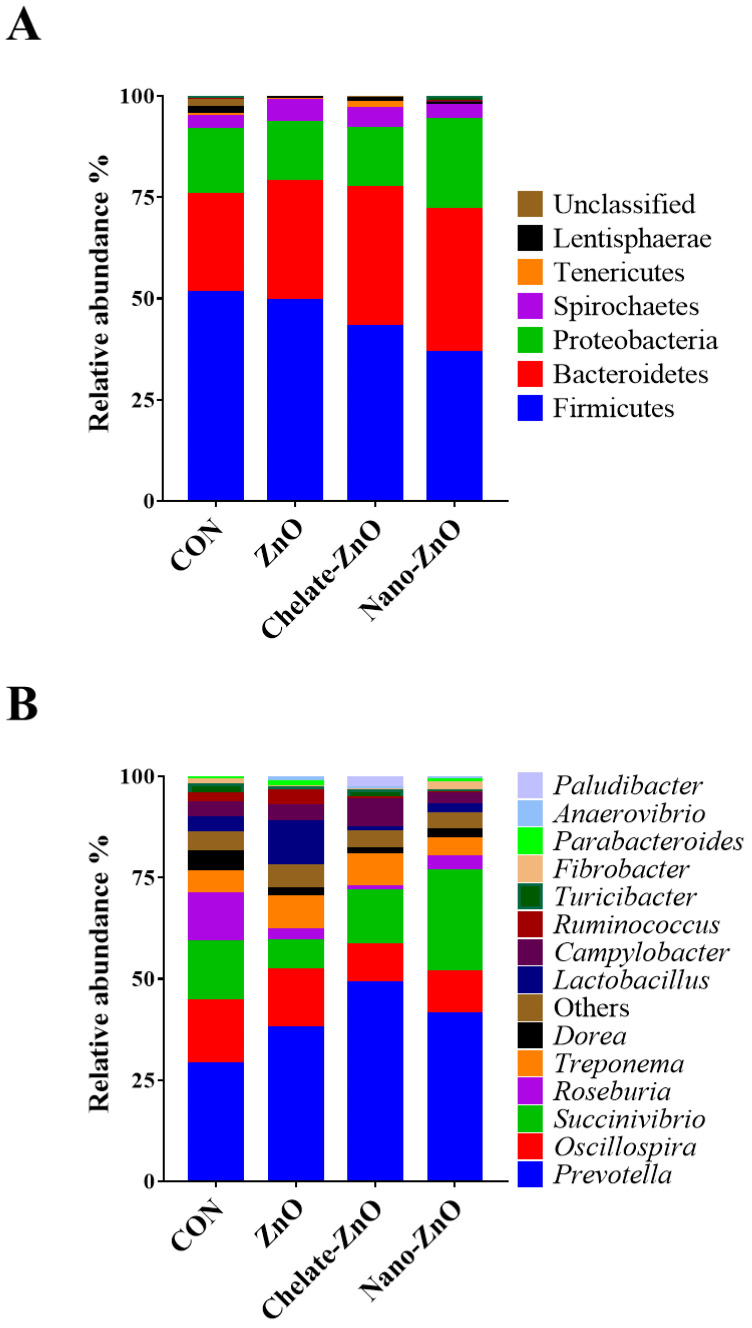
Taxonomic classification of the 16S rRNA gene sequences. Taxonomic analysis at the (**A**) phylum and (**B**) genus levels for the CON, ZnO, chelate-ZnO, and nano-ZnO groups were examined.

**Figure 11 animals-11-01356-f011:**
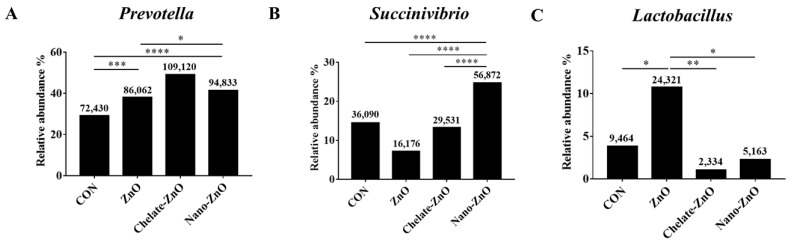
The bar plot identifying the different taxon between control and ZnO treatments. CON, ZnO, chelate-ZnO, and nano-ZnO groups at the genus (**A**–**C**) level were examined. The numbers on each bar indicates the normalized abundance of each strains. * *p*< 0.05; ** *p* < 0.005; *** *p* < 0.001; **** *p* < 0.0001.

**Table 1 animals-11-01356-t001:** Compositions of the basal weaning diets (as-fed basis).

Items	Ingredient, %
Corn	34.43
Extruded corn	15.00
Lactose	10.00
Dehulled soybean meal, 51% CP ^1^	13.50
Soy protein concentrate, 65% CP ^1^	10.00
Plasma powder	6.00
Whey	5.00
Soy oil	2.20
Monocalcium phosphate	1.26
Limestone	1.40
L-Lysine-HCl, 78%	0.06
DL-Methionine, 50%	0.15
Choline chloride, 25%	0.10
Vitamin premix ^2^	0.25
Trace mineral premix ^3^	0.25
Salt	0.40
Total	100.00
Calculated value ^4^	
ME, kcal/kg	3433
CP, %	20.76
Lysine, %	1.35
Metionine, %	0.39
Ca	0.82
P	0.65

^1^ CP, crude protein. ^2^ Provided per kg of complete diet: vitamin A, 11,025 IU; vitamin D_3_, 1103 IU; vitamin E, 44 IU; vitamin K, 4.4 mg; riboflavin, 8.3 mg; niacin, 50 mg; thiamine, 4 mg; d-pantothenic, 29 mg; choline, 166 mg; and vitamin B_12_, 33 mg. ^3^ Provided per kg of complete diet without Zinc: Cu (as CuSO_4_•5H_2_O), 12 mg; Mn (as MnO_2_), 8 mg; I (as KI), 0.28 mg; and Se (as Na_2_SeO_3_•5H_2_O), 0.15 mg. ^4^ Values were calculated using National Swine Nutrition Guide(NSNG; V 2.0).

**Table 2 animals-11-01356-t002:** Primer list in this study.

Primer	Sequence	Size (bp)
GAPDH	Forward: ACATCATCCCTGCTTCTACCGG	126
	Reverse: CTCGGACGCCTGCTTCAC	
TBX21	Forward: TGGACTGAGATCACCCCCAT	103
	Reverse: TGTCCCCACTGGAGGGATAG	
GATA3	Forward: TCTAGCAAATCCAAAAAGTGCAAA	74
	Reverse: GGGTTGAACGAGCTGCTCTT	
RORC	Forward: TTCAGTACGTGGTGGAGTTC	141
	Reverse: TGTGGTTGTCAGCGTTGTAG	
FoxP3	Forward: CGCATGTTCGCCTTCTTCA	68
	Reverse: AGGCTCAAGTTGTGGCGAAT	
IL-1α	Forward: GCAGTGGAGAAGCCGATGAAG	142
	Reverse: GCACGTTGGCATCACAGACA	
IL-2	Forward: ACAGTTGCTTTTGAAGGAAGTTAAGAA	86
	Reverse: CCTGCTTGGGCATGTAAAATTT	
IL-4	Forward: CCAACCCTGGTCTGCTTACTG	119
	Reverse: TCCTTCTCCGTCGTGTTCTCT	
IL-6	Forward: ACAAAGCCACCACCCCTAAC	185
	Reverse: CGTGGACGGCATCAATCTCA	
IL-8	Forward: TGGACCCCAAGGAAAAGTGG	132
	Reverse: TGCAGCAGCAGCTGGAAATTTAT	
IL-10	Forward: TGAGAACAGCTGCATCCACTTC	104
	Reverse: TCTGGTCCTTCGTTTGAAAGAAA	
IL-12β	Forward: TCAGGGACATCATCAAACCA	141
	Reverse: GAACACCAAACATCAGGGAAA	
IL-17A	Forward: CGGCTGGAGAAAGTGATGGT	138
	Reverse: GAAATGGGGCTGGGTCTACTC	
IL-22	Forward: CTGGGAGCCCTTTCCTTCTG	143
	Reverse: GTTGGTGATGTAGGGCTGCT	
IFNα	Forward: CCATTCAAAGGAGCATGGAT	146
	Reverse: GAGTTCACTGATGGCTTTGC	
ZO-1	Forward: AAGGATGTTTACCGTCGCATT	253
	Reverse: ATTGGACACTGGCTAACTGCT	
OCLN	Forward: CAGGTGCACCCTCCAGATTG	167
	Reverse: ATGTCGTTGCTGGGTGCATA	
Claudin 4	Forward: CAACTGCGTGGATGATGAGA	140
	Reverse: CCAGGGGATTGTAGAAGTCG	
pBD1	Forward: CCTGGAAGCAGGAGGTCAAA	194
	Reverse: AAGGGCTATGGATTGTGCGG	
Reg3α	Forward: CCACCGAGGGCTTGGAA	70
	Reverse: GCAACGTAATTGAGCACATCAGA	

**Table 3 animals-11-01356-t003:** Effects of different forms of dietary zinc oxide on the growth performance and diarrhea score of weaned piglets (*n* = 20).

Zn, Status and Level	CON	ZnO	Chelate -ZnO	Nano -ZnO	SE	*p*-Value
Items	0	2500	200	200		Trt
Initial BW, kg	6.4	6.4	6.4	6.4	0.2	0.997
1 week BW, kg	7.5	7.4	7.4	7.4	0.2	0.978
Final BW, kg	8.9	8.9	8.8	8.8	0.2	0.988
0 to 7 days						
ADG, g	153.6	142.9	139.3	142.9	20.2	0.958
ADFI, g	229.6	227.1	245.2	234.0	6.6	0.253
G:F, g/g	0.672	0.632	0.568	0.602	0.083	0.827
Diarrhea score ^1,x^	1.545 ^b^	2.454 ^a^	2.250 ^a^	2.242 ^a^	0.078	0.001
7 to 14 days						
ADG, g	201.4 ^a^	210.7	204.3	204.8	11.0	0.937
ADFI, g	297.5	297.1	300.0	300.0	1.7	0.492
G:F, g/g	0.677	0.710	0.681	0.683	0.038	0.918
Diarrhea score ^1,x,y,z,w^	1.958 ^c^	2.417 ^a^	1.833 ^c^	2.222 ^b^	0.052	0.001
Overall period, 0 to 14 days						
ADG, g	177.5	176.8	171.8	173.8	12.0	0.984
ADFI, g	263.6	262.1	272.6	267.0	3.1	0.126
G:F, g/g	0.674	0.674	0.630	0.649	0.044	0.852
Diarrhea score ^1,x,y^	1.758 ^c^	2.435 ^a^	2.033 ^bc^	2.232 ^ab^	0.076	0.001

Abbreviation: CON, no additional added zinc oxide in diet (negative control); ZnO, CON + 2500 ppm zinc oxide (positive control); Chelate-ZnO, CON + zinc chelate; Nano-ZnO, CON + nanoparticle size of zinc; SE, standard error; BW, body weight; ADG, average daily gain; ADFI, average daily feed intake; G:F, feed efficiency; ^1^ Diarrhea score: Values are calculated by as the average diarrhea score for each period per group by summing the average daily diarrhea scores of each piglets. Diarrhea score was determined as follow: 0, diarrhea; 1, sloppy feces; 2, normal feces; and 3, well-formed feces. ^a,b,^^c^ Means within a row with different letters are significantly different at *p* < 0.05. ^x^ contrast: CON vs. other treatments (*p* < 0.05). ^y^ contrast: ZnO vs. chelate-ZnO (*p* < 0.05). ^z^ contrast: ZnO vs. nano-ZnO (*p* < 0.05). ^w^ contrast: chelate-ZnO vs. nano-ZnO (*p* < 0.05)

**Table 4 animals-11-01356-t004:** Effects of different forms of zinc oxide on the nutrient digestibility of weaned piglets (*n* = 20).

Zn, Status, and Level	CON	ZnO	Chelate -ZnO	Nano -ZnO	SE	*p*-Value
Items	0	2500	200	200		trt
One week intake						
Feed intake, g	232.5	242.5	250.0	250.0	9.5	0.557
Dry matter, g	211.8	220.4	225.5	228.0	8.6	0.611
Crude protein, g	48.3	50.3	52.3	52.1	2.0	0.487
Energy, kcal	914.2	971.9	1016.4	989.0	36.9	0.309
Excretion						
Fresh fecal	220.0	207.5	210.0	213.3	10.6	0.852
Dry matter, g	34.2	30.3	32.6	30.8	1.3	0.225
Crude protein, g	9.5	9.0	9.4	9.4	0.5	0.919
Energy, kcal	184.7	161.1	176.6	163.4	9.8	0.349
One week ATTD, %						
Dry matter ^x^	83.8 ^b^	86.2 ^a^	85.5 ^a^	86.5 ^a^	0.5	0.014
Crude protein	80.3	82.0	82.1	81.9	0.8	0.333
Gross energy ^x^	79.8 ^b^	83.4 ^a^	82.6 ^a^	83.5 ^a^	0.7	0.013
Two weeks intake						
Feed intake, g	300.0	300.0	300.0	300.0	0.0	1.000
Dry matter, g	273.3	272.7	270.6	273.6	0.2	0.977
Crude protein, g	62.3	62.3	62.3	62.3	0.1	0.995
Energy, kcal	1177.5	1203.3	1223.1	1186.8	7.6	0.632
Excretion						
Fresh fecal, g	266.5	282.5	263.8	258.3	8.0	0.254
Dry matter, g	42.8	40.2	42.2	41.0	1.1	0.361
Crude protein, g	12.6	11.2	12.3	11.5	0.4	0.096
Energy, kcal ^y^	237.5 ^ab^	209.1 ^b^	247.1 ^a^	219.5 ^ab^	8.4	0.035
Two week ATTD, %						
Dry matter	84.3	85.3	84.4	85.0	0.4	0.337
Crude protein	79.7	82.1	80.3	81.6	0.7	0.113
Gross energy ^y^	79.8 ^b^	82.6 ^a^	79.8 ^b^	81.5 ^ab^	0.7	0.032

Abbreviation: CON, no additional added zinc oxide in diet (negative control); ZnO, CON + 2500 ppm zinc oxide (positive control); chelate-ZnO, CON + zinc chelate; nano-ZnO, CON + nanoparticle size of zinc; SE, standard error; DE, digestible energy; DM, dry matter; CP, crude protein. ^a,b^ Means within a row with different letters are significantly different at *p* < 0.05. ^x^ contrast: CON vs. other treatments (*p* < 0.05). ^y^ contrast: ZnO vs. chelate- ZnO (*p* < 0.05).

**Table 5 animals-11-01356-t005:** Effects of different forms of zinc oxide on apparent total tract and ileum digestibility in weaned piglets (*n* = 20).

Zn, Status, and Level	CON	ZnO	Chelate -ZnO	Nano -ZnO	SE	*p*-Value
Items	0	2500	200	200		trt
One week						
Feed intake, g	232.5	242.5	250.0	250.0	9.5	0.557
Zinc intake, mg ^x,y,z^	23.3 ^c^	524.6 ^a^	89.0 ^b^	87.3 ^b^	8.0	0.001
Zinc excretion, mg ^x,y,z,w^	21.3 ^d^	461.6 ^a^	36.8 ^c^	69.5 ^b^	4.3	0.001
ATTD of Zinc, % ^x,y,z,w^	8.3 ^c^	11.9 ^c^	58.4 ^a^	20.4 ^b^	2.6	0.001
Two week						
Feed intake, g	300.0	300.0	300.0	300.0	1.7	1.000
Zinc intake, mg ^x,y,z^	30.0 ^c^	663.58 ^a^	106.8 ^b^	107.6 ^b^	2.8	0.001
Zinc excretion, mg ^x,y,z,w^	28.1 ^d^	612.1 ^a^	55.2 ^c^	76.8 ^b^	6.1	0.001
ATTD of Zinc, % ^x,y,z,w^	6.4 ^c^	7.8 ^c^	48.4 ^a^	28.7 ^b^	1.9	0.001
AID of Zinc, % ^x,y,z,w^	5.0 ^c^	5.3 ^c^	39.2 ^a^	23.0 ^b^	0.6	0.001

Abbreviation: CON, no additional added zinc oxide in diet (negative control); ZnO, CON + 2500 ppm zinc oxide(positive control); chelate-ZnO, CON + zinc chelate; nano-ZnO, CON + nanoparticle size of zinc; SE, standard error; ATTD, apparent total tract digestibility; AID, apparent ileum digestibility. ^a,b,c,d^ Means within a row with different letters are significantly different at *p* < 0.05. ^x^ contrast: CON vs. other treatments (*p* < 0.05). ^y^ contrast: ZnO vs. chelate- ZnO (*p* < 0.05). ^z^ contrast: ZnO vs. nano- ZnO (*p* < 0.05). ^w^ contrast: chelate-ZnO vs. nano-ZnO (*p* < 0.05).

**Table 6 animals-11-01356-t006:** Effects of different forms of zinc oxide on the blood profiles of weaned piglets (*n* = 20).

Zn, Status, and Level	CON	ZnO	Chelate -ZnO	Nano -ZnO	SE	*p*-Value
Items	0	2500	200	200		trt
WBC, 10^3^/μL	27.5	20.1	23.5	23.0	2.4	0.253
RBC, 10^6^/μL	7.9	7.8	8.2	7.9	0.3	0.806
Lymphocyte, %	73.4	62.9	62.2	61.5	3.7	0.160
Monocyte, %	0.5	0.6	2.15	0.6	0.4	0.461
Neutrophil, %	23.7	24.7	31.0	37.6	3.0	0.518
IgG, mg/dL	189.0	229.0	191.0	219.0	13.0	0.128
IgM, mg/dL	19.3	20.0	19.3	23.0	2.6	0.663
Zinc, ug/dL ^y,z^	164.9 ^b^	235.5 ^a^	165.3 ^b^	175.8 ^b^	9.6	0.007

Abbreviation: CON, no additional added zinc oxide in diet (negative control); ZnO, CON + 2500 ppm zinc oxide (positive control); chelate-ZnO, CON + zinc chelate; nano-ZnO, CON + nanoparticle size of zinc; SE, standard error; RBC, red blood cell; WBC, white blood cell; IgG, immunogloblin G; IgM, immunogloblin M. ^a,b^ Means within a row with different letters are significantly different at *p* < 0.05. ^y^ contrast: ZnO vs. chelate-ZnO (*p* < 0.05). ^z^ contrast: ZnO vs. nano- ZnO (*p* < 0.05).

## Data Availability

Data sharing is not applicable to this article.

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
