# Peer review of "Changes in Diarrhea Score, Nutrient Digestibility, Zinc Utilization, Intestinal Immune Profiles, and Fecal Microbiome in Weaned Piglets by Different Forms of Zinc"

_animals, 2021, doi:10.3390/ani11051356_

Round 1

Reviewer 1 Report

Oh et al. present an interesting study on the effects of ZnO, nanoparticle-sized ZnO and ZnO chelate with glycine. They show a lot of analysis on only a few animals. This is of course due to the experimental design involving metabolism cages, but still in doing as many analysis, there is always the fact, that there are some significant results per chance. However, I like that they report ALL results, also the ones where they did not get significant results. That is also valuable for the scientific community. But the authors have to be more careful with general conclusions due to the small amount of used animals within each group of their experiment.

L 25: susceptible to various diseases owing to incomplete immune system > better : due to an incomplete IS …

Abstract: Short introduction to the topic is missing. How many piglets per group? Which age, sex  and BW?

Chelate-ZnO (should be more bioavailable) did not improve performance >any explanations?

I would argue that higher IL-6 and IL-8 are not per se better for the animal, since is causes inflammation.

There were higher proportions of T-bet (master transcription factor of PRO inflammatory T helper cells AND at the same time higher proportions for FoxP3+ T helper cells, which are more immune suppressive > Do you have an explanation?

Introduction:

When introducing effects of zinc and ZnO you should also consider the following publication:

Kreuzer-Redmer S, Arends D, Schulte JN, Karweina D, Korkuc P, Wöltje N, Hesse D, Pieper R, Gerdts V, Zentek J, Meurens F, Brockmann GA. High dosage of zinc modulates T-cells in a time-dependent manner within porcine gut-associated lymphatic tissue. Br J Nutr. 2018 Dec;120(12):1349-1358.

Material and Methods:

Animals: Please specify how many animals per group

At which age they were weaned > age of piglets at the experiment?

L 118: “The dietary treatments consisted of CON (negative control; zinc-free diet)” > Hopefully not totally zinc-free. But in corn and Whey will be also some zinc … Additionally you should not induce zinc deficiency in your control fed animals since this would also lead to changed immune response capacity and health status. A good control has no additional added zinc …

L135 – 140: The experimental design is not clearly described. Maybe you can make a little scheme for visualization.

How did you decide from which three piglets you take the feces?

L197: Do not use abbreviations within headers (CBC analysis)

L2013: please correct to : Intestinal tissues were stored in 10% NBF for 2 weeks at room temperature.

L233: please correct: cDNA was synthesized using Accupower RT PreMix …

Results:

L266 Better write: From one to two weeks of age … In general please write numeric till twelve

Table 5: Here you show that you have (fortunately) NOT a zinc free diet in your CON group, since the zinc intake was 23.3/30.0 mg per??? > so please rephrase your table legends

L312 please correct to: Weaned piglets …

Table 1 – 6: n is missing

Figure 1: Why you depict data from 15 piglets and not all 20 piglets used in the experiment?

3.4. > After which time period of feeding did you do the histological analysis? Also consider that two weeks feeding the different diets could be still  considered as a short period of feed supplementation.

Figure 2: The quality has to be improved

L352 and 353: Is already discussion. Additionally, only conclude for your experimental set up, since there are studies which show effects of ZnO on mesenteric lymph node derived lymphocytes (e.g. Janczyk P, Kreuzer S, Assmus J, Nöckler K, Brockmann GA. No protective effects of high-dosage dietary zinc oxide on weaned pigs infected with Salmonella enterica serovar typhimurium DT104. Appl Environ Microbiol. 2013 May;79(9):2914-21.) You included not many samples per group, and as usual for animal experiments, you have high SD. So you may would have seen significant effects with your experimental set up by including more animals.

Figure 3: Please also describe C) and D). Additionally, you should state that you show representative FCM Plots for ONE piglet in A) and C)

L418, please correct the header

Figure 8 would be more informative when you also indicate the ZnO feeding groups e.g. with different coloring; the weeks do not have necessarily be colored since they cluster anyway to Week 2 and Week 0.

At some points its written 2,000 ppm > please check throughout the manuscript

Discussion:

“Additionally, in some studies, zinc supplementation at high dosages leads to negative effects, and alterations are recognized that are similar to those of zinc deficiency [8].” > This effect is probably time dependent …  

L500: “the one-week old control group” ???? > piglets were one week of age at weaning?

L600/601 “However, in our study, numerical values were higher in the Nano-ZnO and ZnO treatments than in the other groups; however, there was no …” > please try to avoid the frequently usage of “however” throughout the whole discussion

L642 > better write: cytokine expression pattern

L671 – 674: “FoxP3+ and T-bet+ T cells were significantly higher in Chelate-ZnO diet-fed pigs

than in ZnO diet-fed pigs. Treg cells (CD4+FoxP3+) limit the development and

progression of the immune response and suppress the development of the inflammatory

response. Th1 cells (CD4+T-bet+) mainly act against viral and bacterial pathogens.” > How you bring your result together? You have a higher portion of ANTI-inflammatory Tregs AND a higher relative cell count of PRO-inflammatory T helper 1 cells > please discuss this fact in more detail  

Conclusion:

“Both chelated ZnO and nanoparticle-sized ZnO showed similar effects, but 2500ppm of ZnO supplementation was excessive in weaning piglets.” > This sentence is not clear to me; please rephrase

“A high dose of zinc oxide could have a negative effect on immune development in the intestine, and a low dosage of zinc oxide could be beneficial to the immune system in weaning piglets.” > This is not a conclusion you can draw from the results of your study

Author Response

Response to Reviewer-1

GENRAL RESPNOSE: Thanks for your all valuable comments. In this study, we tried to provide a variety of biological information, including health status, immune, and growth performance in piglets fed different types of Zinc. To improve a manuscript, we put more efforts, particularly discussion part during revision. We have added more relevant information and implication of our observations from this study. Please, find changes in revised manuscript.

L 25: susceptible to various diseases owing to incomplete immune system > better : due to an incomplete IS …

Response: Thanks for comment. We revised this sentence (Line 25).

Abstract: Short introduction to the topic is missing. How many piglets per group? Which age, sex and BW?

Response: We have added essential information in abstract (Line 39-40).

Chelate-ZnO (should be more bioavailable) did not improve performance >any explanations?

Response: Thanks for comment. We changed contents of abstract (Line 43-44).

I would argue that higher IL-6 and IL-8 are not per se better for the animal, since is causes inflammation.

Response: Thanks for valuable comment. We agree that IL-6 and IL-8 are considered as pro-inflammatory cytokine. If adult animals have unnecessarily high pro-inflammatory cytokines levels, it negatively impacts on growth performance and sometimes induce inflammatory diseases. However, we should not underestimate functional roles of IL-6 and IL-8 in regulation of immunity. For example, IL-6 contributes T cell differentiation and IL-8 involved in many cellular processes, including immune cell proliferation and tissue remodeling. Thus, IL-6 and IL-8 are important cytokines for regulation of immune responses. As we used young animals, which did not have fully developed intestinal immune system, there is possibility that increased IL-6 and IL-8 represents promoted immunity during immune system development. We added more discussions on this issue in revised manuscript (Line 733-742).

There were higher proportions of T-bet (master transcription factor of PRO inflammatory T helper cells AND at the same time higher proportions for FoxP3+ T helper cells, which are more immune suppressive > Do you have an explanation?

Response: Thanks for valuable comment. We agree that the increased numbers or proportion of Th1 cells alongside the increased Treg (regulatory T cells) was not common reaction as they have opposite functional role in regulating immune response. However, there are some reports that immune challenge such as antigen exposure induces robust Th1 or Th17 cell (pro-inflammatory T cells) together with Treg (Ref. Koch et al., 2009; O’Connor et al., 2012). In this response, naïve T cells differentiate Treg cells to suppress inflammation and induce tissue repair process during inflammatory responses. Although, we do not fully understand the development (differentiation) of T cells in the gut of piglets, we assume that intestinal immune system of piglets undergoes robust T cell response particularly, T cell subsets (Th1, Th17, Treg) differentiation from naïve T cells, that make piglets have completed (developed) intestinal immunity. Although, there are limited information on development T cell immunity in piglets, we added more relevant information in discussion part during revision (Line 695-702, 708-709).

Introduction:

When introducing effects of zinc and ZnO you should also consider the following publication:

Kreuzer-Redmer S, Arends D, Schulte JN, Karweina D, Korkuc P, Wöltje N, Hesse D, Pieper R, Gerdts V, Zentek J, Meurens F, Brockmann GA. High dosage of zinc modulates T-cells in a time-dependent manner within porcine gut-associated lymphatic tissue. Br J Nutr. 2018 Dec;120(12):1349-1358.

Response: Thanks for kind comment. We added suggested reference to introduce effect of zinc referring during revision (Line 80-84).

Material and Methods:

Animals: Please specify how many animals per group

At which age they were weaned > age of piglets at the experiment?

Response: Thanks for comment. We added essential information in M&M (Line 123-125).

L 118: “The dietary treatments consisted of CON (negative control; zinc-free diet)” > Hopefully not totally zinc-free. But in corn and Whey will be also some zinc … Additionally you should not induce zinc deficiency in your control fed animals since this would also lead to changed immune response capacity and health status. A good control has no additional added zinc …

Response: Sorry for mistake. We replaced right information in revised manuscript (Line 129).

L135 – 140: The experimental design is not clearly described. Maybe you can make a little scheme for visualization.

Response: Thanks for comment. We added scheme of our experiment in Figure 1.

How did you decide from which three piglets you take the feces?

Response: We randomly selected and collected 3 out of 5 pigs (Line 152-153).

L197: Do not use abbreviations within headers (CBC analysis)

Response: We fixed this. (Line 210).

L213: please correct to : Intestinal tissues were stored in 10% NBF for 2 weeks at room temperature.

Response: We fixed this sentence as followed your comment (Line 226).

L233: please correct: cDNA was synthesized using Accupower RT PreMix …

Response: We corrected this issue (Line 246-247).

Results:

L266 Better write: From one to two weeks of age … In general please write numeric till twelve

Response: Thanks for comment. We revised this sentence (Line 278).

Table 5: Here you show that you have (fortunately) NOT a zinc free diet in your CON group, since the zinc intake was 23.3/30.0 mg per??? > so please rephrase your table legends

Response: Thanks for comment. We revised this sentence (Line 289-290).

L312 please correct to: Weaned piglets …

Response: We corrected this issue (Line 326).

Table 1 – 6: n is missing

Response: We fixed error (Line 288).

Figure 1: Why you depict data from 15 piglets and not all 20 piglets used in the experiment?

Response: Unfortunately, we could not obtain data for spleen size and image from all piglets in this study. We plotted results from 3-4 animals per groups.

3.4. > After which time period of feeding did you do the histological analysis? Also consider that two weeks feeding the different diets could be still considered as a short period of feed supplementation.

Response: Sorry for insufficient information. The intestinal tissue morphology was analyzed after two weeks feeding. We designed experiment with 2 weeks-dietary treatment as followed by study from Bonettie et al., 2021. For weaning piglets, supplementation of high dose ZnO is usually most beneficial for the 14 days after weaning (Bonetti et al). It helps to improve growth performance of weaning piglets by preventing the incidence of PWD and increasing immunity (Ref. 4). Regarding gut tissue morphology, some studies observed different gut tissue structural development by different type or dosage of zinc (reference 55-56). We added relevant information in discussion part (Line 493-496).

REF.: Bonetti, A.; Tugnoli, B.; Piva, A.; Grilli, E. Towards zero zinc oxide: Feeding strategies to manage post-weaning diarrhea in piglets. Animals, 2021, 11, 642.

Figure 2: The quality has to be improved

Response: Thanks for suggestion. We have improved the quality of Figure 2 (currently Figure 3 in revised manuscript) in revised manuscript.

L352 and 353: Is already discussion. Additionally, only conclude for your experimental set up, since there are studies which show effects of ZnO on mesenteric lymph node derived lymphocytes (e.g. Janczyk P, Kreuzer S, Assmus J, Nöckler K, Brockmann GA. No protective effects of high-dosage dietary zinc oxide on weaned pigs infected with Salmonella enterica serovar typhimurium DT104. Appl Environ Microbiol. 2013 May;79(9):2914-21.) You included not many samples per group, and as usual for animal experiments, you have high SD. So you may would have seen significant effects with your experimental set up by including more animals.

Response: We agree with your opinion. One of limitation of our study is shortage of number of samples, as we examined a variety of measurements, including growth performance, health status, gut microbiome, and gut immunity. To strengthen the data from our study, we improved discussion part with more relevant references (Line 713-718).

Figure 3: Please also describe C) and D). Additionally, you should state that you show representative FCM Plots for ONE piglet in A) and C)

Response: Sorry for missing description. The FCM plots in A) and C) were not from one piglet, each of FCM plot were selected most representative samples (Line 371-372).

L418, please correct the header

Response: Thank you for the comment. We have corrected the header as follow (Line 433).

Figure 8 would be more informative when you also indicate the ZnO feeding groups e.g. with different coloring; the weeks do not have necessarily be colored since they cluster anyway to Week 2 and Week 0.

Response: Thanks for suggestion. As suggested, we have indicated each ZnO feeding groups with different colors at Figure 8 (currently Figure 9 in revised manuscript). 

At some points its written 2,000 ppm > please check throughout the manuscript

Response: Thanks for comment. We have checked whole manuscript and corrected errors. 

Discussion:

“Additionally, in some studies, zinc supplementation at high dosages leads to negative effects, and alterations are recognized that are similar to those of zinc deficiency [8].” > This effect is probably time dependent …  

Response: Reference [8] is review paper about in vitro study. We have revised this sentence to provide message more clearly. We also added more references: negative impact on immune response by long term feeding of Zinc from in vivo studies (Line 509-510).

L500: “the one-week old control group” ???? > piglets were one week of age at we aning?

Response: Sorry for error. We fixed this sentence (Line 522).

L600/601 “However, in our study, numerical values were higher in the Nano-ZnO and ZnO treatments than in the other groups; however, there was no …” > please try to avoid the frequently usage of “however” throughout the whole discussion

Response: Thanks for comment. We went through manuscript and tried to delete “however” (Line 621).

L642 > better write: cytokine expression pattern

Response: Thanks for comment. We fixed this sentence (Line 663).

L671 – 674: “FoxP3+ and T-bet+ T cells were significantly higher in Chelate-ZnO diet-fed pigs than in ZnO diet-fed pigs. Treg cells (CD4+FoxP3+) limit the development and progression of the immune response and suppress the development of the inflammatory response. Th1 cells (CD4+T-bet+) mainly act against viral and bacterial pathogens.” > How you bring your result together? You have a higher portion of ANTI-inflammatory Tregs AND a higher relative cell count of PRO-inflammatory T helper 1 cells > please discuss this fact in more detail  

Response: As we already discussed upper part, we think that piglets can have both pro- and anti-inflammatory T cell subsets in the MLN because T cell subset differentiation from naïve T cells happens in intestinal immune system during early age of piglets. We assumed that this is normal T cell response in young piglets. However, we need to precisely examine T cell subsets changes during whole period of piglets in future study. We added relevant discussion in revised manuscript (Line 695-702).

Conclusion:

“Both chelated ZnO and nanoparticle-sized ZnO showed similar effects, but 2500ppm of ZnO supplementation was excessive in weaning piglets.” > This sentence is not clear to me; please rephrase

Response: Sorry for confusing. We revised this sentence to make clear in conclusion part (Line 761).

 “A high dose of zinc oxide could have a negative effect on immune development in the intestine, and a low dosage of zinc oxide could be beneficial to the immune system in weaning piglets.” > This is not a conclusion you can draw from the results of your study.

Response: Thanks for comment. We have deleted these contents make clear on conclusion of our study (Line 761-765).

Reviewer 2 Report

In this study, the authors investigated the effect of various forms of zinc on diarrhea score, nutrient digestibility, Zinc utilization, intestinal immune profiles, and fecal microbiome in weaned piglets, and suggested that nanoparticle-sized ZnO could be used as a ZnO alternative. The study provided a very large body of results and provides us with new information, especially in the field of adaptive immune response.

However, there are some specific issues to be addressed:

1) The diet contained 2500 mg of zinc oxide (ZnO) per kg diet. According to previous studies, this high dose of ZnO acts as an endurance promoter, which improves the body weight gain (BW) and average daily gain (ADG) of weaned piglets, what about the effect of nanoparticle-sized ZnO on ADG of weaned piglets compared to CON diet or 2500 ppm ZnO in this study? Difference in growth performance among the four treatments should be added to the manuscript.

2) In this study, the author analyzed the fecal microbiome, I wonder if the feces were sampled freshly for this analysis? If not, digesta from caecum or colon may be more desirable to evaluate the relationship between diet and gut microbiota.

3) The bioinformatics analysis of gut microbiota should be provided.

4)The authors determined the adaptive immune response in the mesenteric lymph node (MLN) but not in the small intestine of weaned piglets. Did the alternative form of ZnO-induced differentiation of CD4+ T cells in the MLN have any effects on diarrhea? Please discuss.

5) In conclusion, nanoparticle-sized ZnO could enhance beneficial genera that contribute to the degradation of polysaccharides and complex carbohydrates of plant cell walls into SCFAs. Did you investigate the levels of short chain fatty acids, especially butyrate in the colonic digesta or the fresh fecal samples?

6) Line 197, what are CBC?

7) Line 40-43 The dietary treatments included a negative control (CON), standard ZnO (ZnO, 2,000ppm), zinc chelate with glycine (Chelate-ZnO, 200ppm), and nanoparticle-sized ZnO (Nano-ZnO, 200ppm), but in Line 118-120: The dietary treatments consisted of CON (negative control; zinc-free diet), ZnO (positive control; CON + 2,500 ppm/kg zinc oxide), Chelate-ZnO (CON + 200 ppm/kg of zinc chelate with glycine), and Nano-ZnO (CON + 200 ppm/kg nanoparticle-sized zinc oxide; Table 1).Please confirm the amount of standard ZnO.

Author Response

Response to Reviewer-2

1) The diet contained 2500 mg of zinc oxide (ZnO) per kg diet. According to previous studies, this high dose of ZnO acts as an endurance promoter, which improves the body weight gain (BW) and average daily gain (ADG) of weaned piglets, what about the effect of nanoparticle-sized ZnO on ADG of weaned piglets compared to CON diet or 2500 ppm ZnO in this study? Difference in growth performance among the four treatments should be added to the manuscript.

Response: Thank you for your consideration. We added growth performance data (Table 3) and revised result in paper.

2) In this study, the author analyzed the fecal microbiome, I wonder if the feces were sampled freshly for this analysis? If not, digesta from caecum or colon may be more desirable to evaluate the relationship between diet and gut microbiota.

Response: Thanks for your comment. We conducted the longitudinal analyses of the pig gut microbiome. Therefore, we followed the same pigs during the experimental period, and collected fecal samples directly from the rectum of the animals (Line 151-152).

3) The bioinformatics analysis of gut microbiota should be provided.

Response: Thanks for the comment. The bioinformatics analysis of gut microbiota has been described in L184-197.

4)The authors determined the adaptive immune response in the mesenteric lymph node (MLN) but not in the small intestine of weaned piglets. Did the alternative form of ZnO-induced differentiation of CD4+ T cells in the MLN have any effects on diarrhea? Please discuss.

Response: As we described in original version of manuscript, the MLN is major tissue site that naïve T cell were differentiated into T cell subsets in response to luminal antigens presented by antigen presenting cells (APCs). Differentiated T cells migrate to small or large intestine to perform their immune function at effector site. Thus, MLN is the most important area for T cell immune response in the gut. If piglet have pathogen infection, which induce diarrhea, naïve T cells differentiate into responding T cells (Th1 or Th17) in MLN then migrate into intestinal tissue to fight against pathogen. This is normal mucosal immunity against pathogens. We added more explanation for this (Line 679-680)

5) In conclusion, nanoparticle-sized ZnO could enhance beneficial genera that contribute to the degradation of polysaccharides and complex carbohydrates of plant cell walls into SCFAs. Did you investigate the levels of short chain fatty acids, especially butyrate in the colonic digesta or the fresh fecal samples?

Response: Thank you very much for the comment. We totally agree with reviewer’s comment. Although, we did not directly examine SCFA concentration, please understand that we analyzed 16S rRNA genes and referred to reference papers to explain the functions of each genera. We cited reference papers for the functions of each genera as follows.

L543-550: Prevotella 26, 47, and 48; Succinivibrio 49; Lactobacillus 50, 51, and 52

6) Line 197, what are CBC?

Response: Sorry for insufficient information. CBC is an abbreviation of complete blood count, we changed CBC to full name (Line 212).

7) Line 40-43 The dietary treatments included a negative control (CON), standard ZnO (ZnO, 2,000ppm), zinc chelate with glycine (Chelate-ZnO, 200ppm), and nanoparticle-sized ZnO (Nano-ZnO, 200ppm), but in Line 118-120: The dietary treatments consisted of CON (negative control; zinc-free diet), ZnO (positive control; CON + 2,500 ppm/kg zinc oxide), Chelate-ZnO (CON + 200 ppm/kg of zinc chelate with glycine), and Nano-ZnO (CON + 200 ppm/kg nanoparticle-sized zinc oxide; Table 1).Please confirm the amount of standard ZnO.

Response: We confirmed that it was 2,500ppm. Sorry for our mistake.

Reviewer 3 Report

The submitted manuscript, Animals 2021, entitled " Changes in Diarrhea Score, Nutrient Digestibility, Zinc Utilization, Intestinal Immune Profiles, and Fecal Microbiome in Weaned Piglets by Various Forms of Zinc" is a good paper by showing the effects of various forms of ZnO alternatives on piglets.The conclusions are very useful to search a promising alternative to ZnO for piglets.

After reading carefully through the paper, some comments are kindly recommended as followed:

Specific comments:

1.You should clarify how the dosage of Chelate-ZnO and Nano-ZnO were determined

2.Line284-286, “The energy excretion in the feces of the ZnO treatment group was significantly decreased compared with that of the chelate treatment group aged between 1 to 2 weeks”.The sentence does not agree with the table 4

Author Response

Response to Reviewer-3

Specific comments:

1.You should clarify how the dosage of Chelate-ZnO and Nano-ZnO were determined

Response: Thank you for your comment. We added what you said to the introduction, citing the Council (2012) (Line 110-111).

2.Line284-286, “The energy excretion in the feces of the ZnO treatment group was significantly decreased compared with that of the chelate treatment group aged between 1 to 2 weeks”. The sentence does not agree with the table 4

Response: Thank you for your consideration. We revised ‘aged between 1 to 2 weeks’ to ‘in 2 weeks’ (Line 320).

Round 2

Reviewer 2 Report

The manuscript has been improved, but the full text still needs to be carefully checked and revise.

In the expression of “various forms of dietary zinc oxide”, “various” is suggested to be replaced by “different”

Line 39-40: Twenty weaned piglets, five piglets per group, (initial body weight: 6.83±0.33kg; 21 day 39 of age; LYD) were used in a 2-week feeding trial to determine the effects of various dietary zinc 40 forms.

Suggested version: Twenty weaned piglets with initial body weight of 6.83±0.33kg (21 day of age, LYD) were randomly assigned to 4 treatments for a 2-week feeding trial to determine the effects of different dietary zinc on intestinal health of piglets.

Line 64: mixing of pigs from different farms. Weaning piglets are typically mixed among pens rather than farms.

Examples of phases that are confusing are showed below:

Line 278  At 0 to 1 week of after weaning

Line 281  At 1 to 2 weeks of after weaning

Line 286  there were no significant differences

Line 522  the one-week after weaning control group

Line 762  or

Incomplete sentence, A total of 123 20 weaned piglets[(Yorkshire×Landrace) ×Duroc; average initial body weight of 6.43 ± 124 0.33 kg; 21 day of age]. (line 123-124)

Duplicate sentences, Line 123-126: A total of 123 20 weaned piglets[(Yorkshire×Landrace) ×Duroc; average initial body weight of 6.43 ± 124 0.33 kg; 21 day of age]. The pigs (average initial body weight of 6.43 ± 0.33 kg) were 125 individually placed in 45 cm × 55 cm × 45 cm stainless steel metabolism cages in an 126 environmentally controlled room (30 ± 1 °C).

punctuation marks also need to be checked, for example,

line 39  five piglets per group, (initial body weight: 6.83±0.33kg; 21 day 39 of age; LYD)

line 45   scores but, not Chelate-ZnO compared

Author Response

Response to reviewer 2 (round 2)

The manuscript has been improved, but the full text still needs to be carefully checked and revise.

General Response: We really appreciate all your valuable comments. We have corrected our manuscript according to your kind suggestions. In addition, we went through whole text and further revised to improve our manuscript during second round of revision. If we need to have any helps from English editing service during editorial process, we would to like to get it to improve it. Thanks again for your all efforts.

In the expression of “various forms of dietary zinc oxide”, “various” is suggested to be replaced by “different”

Response: Thank you for kind suggestion. I have replaced ‘various’ by ‘different’ in whole text.

Line 39-40: Twenty weaned piglets, five piglets per group, (initial body weight: 6.83±0.33kg; 21 day 39 of age; LYD) were used in a 2-week feeding trial to determine the effects of various dietary zinc 40 forms.

Suggested version: Twenty weaned piglets with initial body weight of 6.83±0.33kg (21 day of age, LYD) were randomly assigned to 4 treatments for a 2-week feeding trial to determine the effects of different dietary zinc on intestinal health of piglets.

Line 64: mixing of pigs from different farms. Weaning piglets are typically mixed among pens rather than farms.

Response: Thanks for kind suggestion. We have revised this sentence as followed your suggestion

(Line 65).

Examples of phases that are confusing are showed below:

Line 278  At 0 to 1 week of after weaning

Line 281  At 1 to 2 weeks of after weaning

Line 522  the one-week after weaning control group

Response: Thanks for comment. We revised these sentences to make clearer.

  • ‘At 0 to 1 week of after weaning’ to ‘at 0 to 7 days’ (Line 280).
  • ‘At 1 to 2 week of after weaning’ to ‘at 7 to 14 days’ (Line 283).
  • ‘the one-week after weaning control group’ to ‘than pigs fed with control group at one week.’ (Line 525).

Line 286  there were no significant differences

Line 762  or

Response: Thank you for comment. We have deleted or revised these sentence during revision.

Incomplete sentence, A total of 123 20 weaned piglets[(Yorkshire×Landrace) ×Duroc; average initial body weight of 6.43 ± 124 0.33 kg; 21 day of age]. (line 123-124)

Duplicate sentences, Line 123-126: A total of 123 20 weaned piglets[(Yorkshire×Landrace) ×Duroc; average initial body weight of 6.43 ± 124 0.33 kg; 21 day of age]. The pigs (average initial body weight of 6.43 ± 0.33 kg) were 125 individually placed in 45 cm × 55 cm × 45 cm stainless steel metabolism cages in an 126 environmentally controlled room (30 ± 1 °C).

Response: Thank you for comments. We corrected these sentences. (Line 123-127).

punctuation marks also need to be checked, for example,

line 39  five piglets per group, (initial body weight: 6.83±0.33kg; 21 day 39 of age; LYD)

Response: Thanks for comment. We revised that (Line 39-41).

line 45   scores but, not Chelate-ZnO compared

Response: Thanks for comment. We revised it. (Line 46).